# The R203M and D377Y mutations of the nucleocapsid protein promote SARS-CoV-2 infectivity by impairing RIG-I-mediated antiviral signaling

Yongkui Li[1,2,3ꚍ], Moran Li[2,3ꚍ], Heng Xiao[2,3ꚍ], Feng Liao[4ꚍ], Miaomiao Shen[5ꚍ], Weiwei Ge[2,3], Junxian Ou[2,3], Yuqing Liu[2,3], Lumiao Chen[6], Yue Zhao[2,3], Pin Wan[2,3], Jinbiao Liu[2,3], Jun Chen[2,3], Xianwu Lan[6], Shaorong Wu[6], Qiang Ding[7], Geng Li[4*], Qiwei Zhang[2,3*], Pan Pan [1,2,3,6*]

1 State Key Laboratory of Respiratory Disease, School of Basic Medical Science, Guangzhou Medical University, Guangzhou, China, 2 Key Laboratory of Viral Pathogenesis & Infection Prevention and Control (Jinan University), Ministry of Education, Guangzhou, China, 3 Department of Immunology and Microbiology, Institute of Medical Microbiology, College of Life Science and Technology, Jinan University, Guangzhou, China, 4 Laboratory Animal Center, Guangzhou University of Chinese Medicine, Guangzhou, China, 5 Guangzhou Institutes of Biomedicine and Health, Chinese Academy of Sciences, Guangzhou, China, 6 The First Affiliated Hospital of Jinan University, Guangzhou, China, 7 School of Medicine, Tsinghua University, Beijing, China

ꚍ These authors contributed equally to this work.
* panpan@gzhmu.edu.cn (PP); lg@gzucm.edu.cn (GL); zhangqw@jnu.edu.cn (QZ)

## Abstract

The viral protein mutations can modify virus-host interactions during virus evolution, and thus alter the extent of infection or pathogenicity. Studies indicate that nucleocapsid (N) protein of SARS-CoV-2 participates in viral genome assembly, intracellular signal regulation and immune interference. However, its biological function in viral evolution is not well understood. SARS-CoV-2 N protein mutations were analyzed in Delta, Omicron, and original strains. Two mutations with a methionine (M) residue at site 203 and a tyrosine (Y) residue at site 377 of the N protein were found in Delta strain but not in Omicron and original strains, and pro-moted SARS-CoV-2 infection therein. Those mutations, R203M and D377Y, enhanced the inhibitory impact of N protein on the impairment of RIG-I-mediated antiviral signaling, such as IRF3 phosphorylation and IFN-β activation. The viral RNA-binding activity of N protein was promoted by these mutations, effectively attenuating the recognition and interaction of RIG-I with viral RNA compared to the original or other variants. The R203M/D377Y muta-tions thus enhanced the suppressive activity of the N protein on RIG-I-mediated interferon induction both in vitro and in vivo, which in turn promoted viral replication. This study helps to understand the variability of SARS-CoV-2 in regulating host immunity.

## Author summary

Recently, the World Health Organization declared that the COVID-19 pandemic is no longer a Public Health Emergency of International Concern (PHEIC). However, the

**Data availability statement:** All relevant data are within the manuscript and its Supporting Information files.

**Funding:** This work was supported by the National Natural Science Foundation of China (32200117 to P.P. and 92269103 to Q.Z.), Open Research Fund Program of the State Key Laboratory of Virology of China (2021KF003 to P.P.), Open Research Fund Program of Guangdong Provincial Key Laboratory of Virology (2022KF003 to P.P.), R&D Program of Guangzhou Laboratory (SRPG22-006 to Q.Z.). Guangdong Basic and Applied Basic Research Foundation (2024A1515011433 to Y.L., 2021A1515011272 to J.L., and 2024A1515013063 to P.P), and Fundamental Research Funds for the Central Universities (21623404 to Y.L. and 21623222 to P.P) Guangzhou Science and Technology Plan Project (Youth Doctor "Setting Sail", 2024A04J4102 to P.P). The funders had no role in study design, data collection and analysis, decision to publish, or preparation of the manuscript.

**Competing interests:** The authors have declared that no competing interests exist.

SARS-CoV-2 virus continues to infect and transmit, with ongoing mutations leading to the emergence of multiple variants that still pose a threat to human health. Among these, compared to the original strain, the Delta variant demonstrates faster transmission speed, higher viral load and a higher likelihood of causing severe disease. In contrast, the Omicron variant and its sublineages (including XBB, BQ.1.1, and CH.1.1) exhibit weaker replication ability but possess stronger immune escape and transmission capabilities. Therefore, closely monitoring different mutant strains of SARS-CoV-2 and investigating the impact of mutations on the virus is of significant scientific importance. The N protein of SARS-CoV-2 is closely related to viral replication. We have found that the R203M and D377Y mutations in the N protein of the Delta variant can competitively bind to viral RNA with RIG-I, thereby inhibiting interferon production regulated by the RIG-I-MAVS signaling pathway and promoting viral replication. Our study elucidates the biological function of the N protein during the evolution of SARS-CoV-2 and provides new insights into the regulatory pathways of viral replication in the virus's evolutionary process.

## Introduction

Since the COVID-19 outbreak, the SARS-CoV-2 pandemic has persisted for over 3 years, resulting in severe respiratory failure or even death in critical cases [1,2]. New viral variants have emerged during this extensive pandemic [3,4]. The World Health Organization (WHO) has categorized these variants based on their transmissibility and virulence. They are classified as the variant strains of concern (VOC) [5,6], which include Alpha (No. B.1.1.7), Beta (No. B.1.351), Gamma (No. B.1.1.28.1), Delta (No. B.1.617.2) and Omicron (No. B.1.1.529), as well as the variant strains of interest (VOI), including Eta (No. B.1.525), Iota (No. B.1.526), Kappa (No. B.1.617.1), and Lambda (No. C.37) [7,8]. The Delta variant is highly infectious, with a greater capacity for infection and replication than the original strain [9,10]. Compared to the Delta strain, Omicron, the current predominant variant, has a lower replication potential in the host but a greater capacity for immune escape and transmission [11,12]. Therefore, it is crucial to monitor SARS-CoV-2 variants and study the impact of mutation site changes on virus.

Spike proteins mediate SARS-CoV-2 entry into cells by binding to receptors on the surface of human cells. They are also the target molecules for SARS-CoV-2 vaccine design and specific neutralizing antibody therapies [13,14]. Mutations in the spike protein alter the characteristics of viral invasion into cells and reduce the protection provided by neutralizing antibodies induced by vaccines. Consequently, specific neutralizing antibody therapies have become less effective. Previous studies have focused on spike protein mutations. For example, the D614G mutation increase viral fitness, infectivity and fatality [15,16]; the N501Y mutation enhance viral infectivity and resistance to neutralizing antibodies [17]; the E484K mutation allows the virus to evade most monoclonal antibodies [18], etc. SARS-CoV-2 has a complex genome that also includes several open reading frames (ORFs) with key roles in viral pathogenesis. Notably, ORF3 [19], ORF6 [20] and ORF9b [21] are critical for antagonizing the host's innate immune response. These ORFs interfere with the host's antiviral defenses by disrupting interferon signaling and other immune mechanisms, thereby facilitating viral persistence and replication. Among these, ORF6 has been identified as crucial for viral replication. Particularly noteworthy is the D61L mutation in ORF6, which emerged in the Omicron strains. This mutation impairs immune evasion, highlighting its essential role in the virus's ability to propagate [20]. Therefore, it is crucial to investigate

mutations in other regions of the SARS-CoV-2 genome that alter the virus's virulence and transmissibility.

The nucleocapsid protein (N), another structural protein of SARS-CoV-2, plays a significant role in the viral life cycle [22]. The N protein promotes its own replication by inhibiting the interferon pathway [23,24] or suppressing the formation of cellular stress granules (SG) [25,26]. It also activates NLRP3 inflammasome through directly interacting with NLRP3, leading to an inflammatory storm in the lungs [27]. The amino acid sequence of SARS-CoV-2 N protein is conserved across variants, however there are mutations at important sites. Mutations at amino acids 203 (R mutated to K) and 204 (G mutated to R) of the N protein enhance its replication [28,29]. These two mutations were first found in the Alpha, Gamma, and Omicron variants, but not in the N protein of Delta variant [30]. Delta has three major mutation sites in N protein, i.e., position 63 (D mutated to G), 203 (R mutated to M) and 377 (D mutated to Y). Omicron variant N protein does not contain these mutations as present in Delta, except for the aforementioned mutations at amino acids 203 (R mutated to K) and 204 (G mutated to R), and another at position 13 (P mutated to L). Further investigation is needed to determine whether these mutations are linked to the virulence of the Delta and Omicron variants.

In the present study, the impact of mutant sites in the N protein of Delta and Omicron variants on replication was screened using a viral-like particle system (N gene of SARS-CoV-2 genome was replaced by green fluorescent protein. The packaging and maturation of virus was completed in cells with stably expressed N protein. The system could be operated in P2 laboratory) and further explored with live SARS-CoV-2 in P3 laboratory [31]. Results exhibited that the Delta N protein with mutations at sites 203 and 377 promoted viral replication compared to original N protein. In contrast, mutations at 13 and 203/204 of Omicron N protein slightly inhibited the viral replication compared to the original N protein. The mechanism behind this involved the Delta N protein mutations at sites 203 and 377 inhibiting the bindings of RIG-I and MAVS, thereby inhibited TBK1 and IRF-3 phosphorylation, preventing IRF-3 entry into the nucleus, and reducing IFN-β production. Delta N protein had mutations at sites 203 and 377 that promoted binding to viral self RNA and inhibited RIG-I ability to bind SARS-COV-2 RNA. It is noteworthy that, compared to the original N protein, mutations at the 13 and 203/204 sites in the Omicron N protein had little effect on interferon expression. However, these mutations inhibited viral replication by suppressing the binding of the N protein to viral RNA. Thus, the N protein mutations in SARS-CoV-2 variants affected viral infection through different mutation sites.

## Results

### N protein mutations in SARS-CoV-2 variants

The amino acid sequence of SARS-CoV-2 N protein is generally conserved; however, it differs among variants [30]. A phylogenetic tree constructed with the N protein revealed that SARS-CoV-2 strains cluster separately from murine coronavirus strains and are divided into three branches; Delta strains clustered into one branch with Omicron, Lambda, Alpha and Gamma strains, while Beta and Mu strains clustered into one branch, and wild-type original virus strains into single branch (S1A Fig). Next, the positive selection analysis of genes was performed. Model 0 and Two ratio Model 2 were compared in the branch model. Model 0 assumed that all branches in evolutionary tree had same ω value, while Two ratio Model 2 assumed foreground branches having different ω values than of background branches. The model comparison resulted in P < 0.01 and Delta strain as foreground branch ω 2 > 1, which reflected that Delta N protein was less constrained with other SARS-COV-2 variants and there

might be positive selection (Table 1). The amino acid sequence comparison exhibited that compared with original N protein, Delta variant was mutated from D to G at position 63, R to M at 203, and D to Y at 377. Omicron was mutated from R to L at position 13, R to K at 203, and G to R at 204 (Fig 1A). The structure of SARS-CoV-2 Delta (Fig 1B) or Omicron (Fig 1C) variant N protein were reconstructed using SWISS-MODEL Server based on original N protein model. The structures of those two mutant proteins were relatively conserved, with D63G located in the NTD, R203M located in the linkage region and D377Y located in the C-terminus of the Delta mutant N protein, while Omicron variant including P13L located in the N-terminus and R203K, G204R mutation located in linkage region. Temporal evolution of above mutated N protein analysis by using web (www.nextstrain/ncov) found that D63G, R203M, and D377Y appeared in Delta (Fig 1D), P13L in Omicron, while R203K and G204R appeared early but did not appear in Delta, and continued to appear in subsequent Omicron (Fig 1E). N protein clones of Delta and Omicron variants were constructed. N protein clones were transfected in HEK293T cells (S1B and S1C Fig) or Caco-2 cells (S1D and S1E Fig). Results revealed that N proteins with different point mutations were expressed in the cells, and they had no effect on mRNA and protein levels of N proteins.

## R203M/D377Y mutation of N protein promoting SARS-CoV-2 infection

Previous study found that mutations at 203 and 204 sites of SARS-CoV-2 N protein promoted viral replication [28,29]. The effect of other point mutations in N protein was investigated regarding its own replication. A previously reported system was employed, i.e., SARS-CoV-2-trVLP, wherein N gene was replaced by green fluorescent protein (GFP). It was necessary to have cells with stably expressed N protein for completing the viral packaging and maturation. This system could be operated in P2 laboratory [31]. Thus, the effect of mutant N proteins regarding own viral replication could be explored by this system. HEK293T cells were easy to transfect and they were not susceptible to SARS-CoV-2 infection. HEK293T cells stably expressing ACE2 (HEK293T-ACE2) were constructed using lentiviral vector system (S2A Fig). Caco-2 cells expressing original SARS-CoV-2 N protein (Caco2-N) were also constructed using lentiviral vector system (S2B Fig). The procedure was as follows: HEK293T-ACE2 cells (S2C Fig) were transfected with original and mutant N protein, respectively, and then infected with SARS-CoV-2-trVLP after 24 hours of transfection. GFP expression level was observed by fluorescence microscopy after 24 hours of infection with trVLP (S2D Fig). The transfected protein expression in cells was observed by WB and virus in cell culture supernatant by absolute quantitative PCR (S2E Fig). The trVLP in supernatant continued to infect Caco-2-N

**Table 1. Results of branch model.**

| Model | np | Ln L | Estimates of parameters | Model compared | LRT P-value | Omega for Branch |
|---|---|---|---|---|---|---|
| Two ratio Model 2 | 17 | −2029.429 | $\omega_1 = 0.160$ $\omega_2 = 64.417$ | Model 0 vs. Two ratio Model 2 | 0.003 | 64.417 |
| Model 0 | 16 | −2033.915 | $\omega = 0.199$ | | | |

$\omega_0$ is the value of all branches predicted by Model 0; $\omega_1$ is the value of background branch and $\omega_2$ is the value of foreground branch under the prediction of Two ratio Model 2. $\omega > 1$ represents positive selection, $\omega < 1$ denotes purifying selection, whereas $\omega = 1$ denotes neutral or natural selection. In branch model, Model 0 and Two ratio Model 2 are mainly compared. Model 0 assumes that all branches in the evolutionary tree have the same $\omega$-values, while the Two ratio Model 2 assumes that foreground and background branches have different $\omega$ values. The results of Table 1 showed that the results of model comparison were P < 0.01 and delta was the foreground branch $\omega_2 > 1$, indicating that N protein of delta is less constrained in natural environment than other SARS-COV-2 strains, and there may be positive selection.

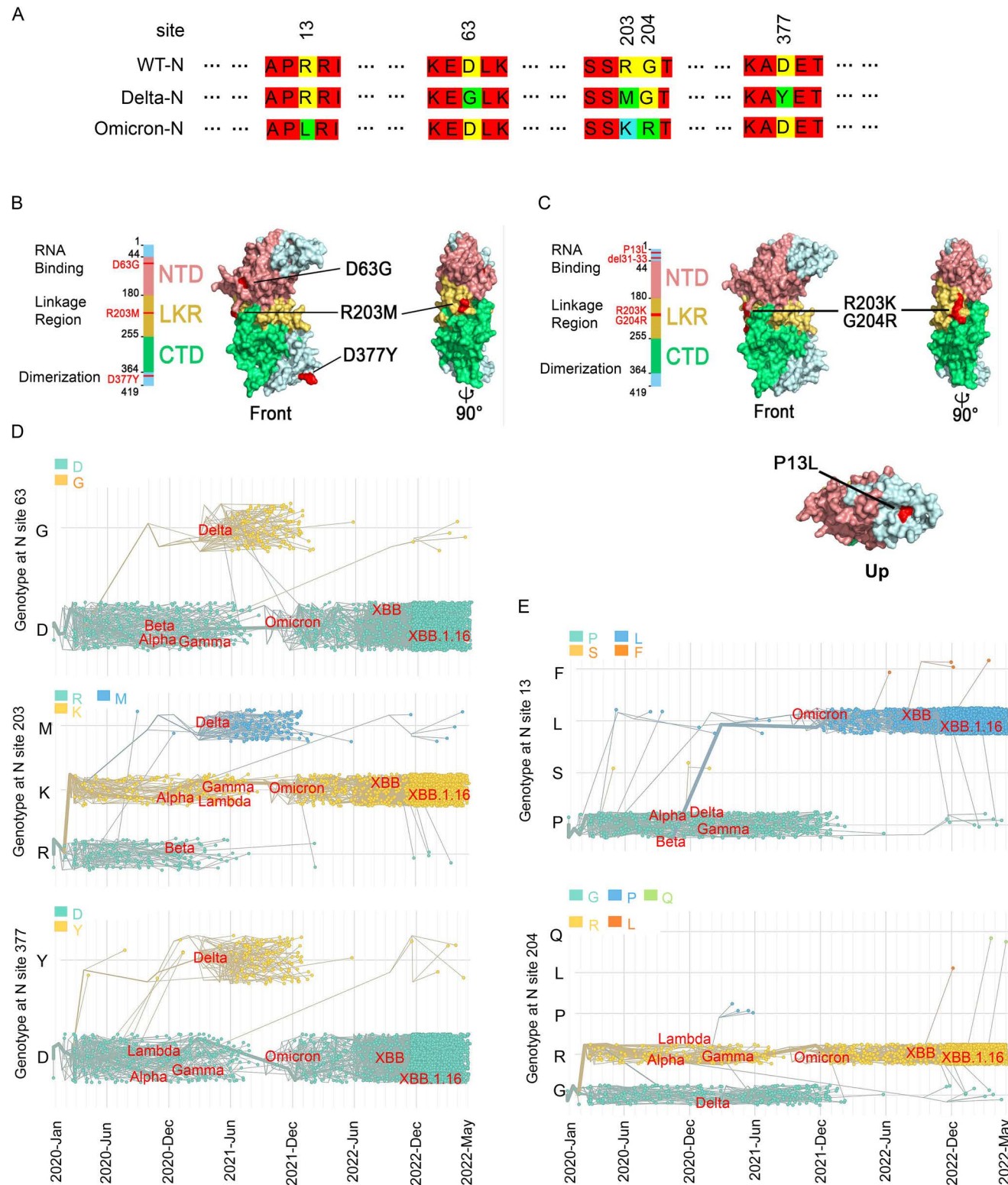

**Fig 1. Amino acid sequence alignment and temporal evolution of main site mutation of N proteins. (A)** Amino acid sequence alignment among original N protein (WT-N), Delta variant N protein (Delta-N) and Omicron variant N protein (Omicron-N). **(B, C)** The structure of SARS-CoV-2 Delta (B) and Omicron (C) variant N protein as shown, mutated residue D63G, R203M and D377Y of Delta variant, and P13L, R203K and G204R of Omicron variant N protein were annotated in red color, functional regions NTD, LKR and CTD of N protein were annotated in different colors, separately. **(D)** Temporal evolution of Delta N protein at position 63 (A), 203 (B) and 377 (C). **(E)** Temporal evolution of Omicron N protein at position 13 (A) and 204 (C). All the temporal evolution of above mutated N protein analysis by using web (www.nextstrain/ncov).

cells for verifying the infection level of live virus. GFP expression level in Caco2-N cells was observed by fluorescence microscopy after 24 hours of infection (S2F Fig). The copies of virus in supernatant were measured by absolute quantitative PCR (S2G Fig). Results depicted that: compared with original N protein, Delta N protein having mutations containing sites R203M and D377Y promoted trVLP replication; while the Omicron mutant N protein (containing sites P13L and RG203/204KR) reduced promotion of trVLP replication as compared to Delta mutant N protein (containing sites D63G, R203M and D377Y) or original N protein (S2D–S2G Fig).

Next, we investigated the effects of different mutated N proteins on SARS-CoV-2-trVLP replication using Caco-2 cells. The procedure was as follows: Caco-2 cells (Fig 2A) were transfected with original and mutant N protein, respectively, after transfection for 24 hours, infected with SARS-CoV-2-trVLP for another 24 hours. GFP expression level was observed by fluorescence microscopy after 24 hours of infection with trVLP (Fig 2B). The transfected protein expression in cells was observed by WB and virus in cell culture supernatant by absolute quantitative PCR (Fig 2C). The trVLP in supernatant continued to infect Caco-2-N cells for verifying the infection level of live virus. GFP expression level in Caco2-N cells was observed by fluorescence microscopy after 24 hours of infection (Fig 2D). The copies of virus in supernatant were measured by absolute quantitative PCR (Fig 2E). Results depicted that: compared with original N protein, Delta N protein having mutations containing sites R203M and D377Y promoted trVLP replication; while the Omicron mutant N protein (containing sites P13L and RG203/204KR) reduced promotion of trVLP replication as compared to Delta mutant N protein (containing sites D63G, R203M and D377Y) or original N protein (Fig 2B–2E).

Finally, we further use the original SARS-CoV-2 to prove that the mutation of N protein R203M and D377Y can promote its replication. The procedure was as follows: Caco-2 cells (Fig 3A) were transfected with original and mutant N protein, respectively, and then infected with original SARS-CoV-2 after 24 hours of transfection. The virus in cell by relative quantitative PCR and (Fig 3B) in cell culture supernatant by absolute quantitative PCR (Fig 3C). The SARS-CoV-2 in supernatant continued to infect Caco-2 cells for verifying the infection level of live virus. The virus in cell by relative quantitative PCR after 24 hours of infection (Fig 3D). The copies of virus in supernatant were measured by absolute quantitative PCR (Fig 3E). Results were similar with the trVLP above: Delta N protein having mutations containing sites R203M and D377Y promoted SARS-CoV-2 replication more strongly than the Omicron mutant N protein (containing sites P13L and RG203/204KR) and original N protein; in fact, the mutations P13L and RG203/204KR in the Omicron mutant N protein reduced its activity on promoting viral replication compared with original N protein.

## R203M/D377Y mutation enhancing inhibitory role of N protein in IFN-β induction

Studies had found that SARS-CoV-2 N protein inhibited interferon production [24,25]. We first compared the viral growth kinetic and interferon changes of the original strain, Delta variant, and Omicron variant in Caco-2 cells at various time points post-infection (e.g., 6, 12, 24, 48 and 72 hours). The experimental results showed that compared to the original strain, the Delta variant exhibited significantly enhanced replication capability starting from 12 hours post-infection (Fig 4A), which corresponded to a significant decrease in the level of interferon produced by the infected cells (Fig 4B). In contrast, compared to both the original strain and the Delta variant, the Omicron variant demonstrated a significant reduction in replication capability starting from 6 hours post-infection, accompanied by a significant increase in the level of interferon produced by the infected cells (Fig 4A and 4B). These findings suggest that

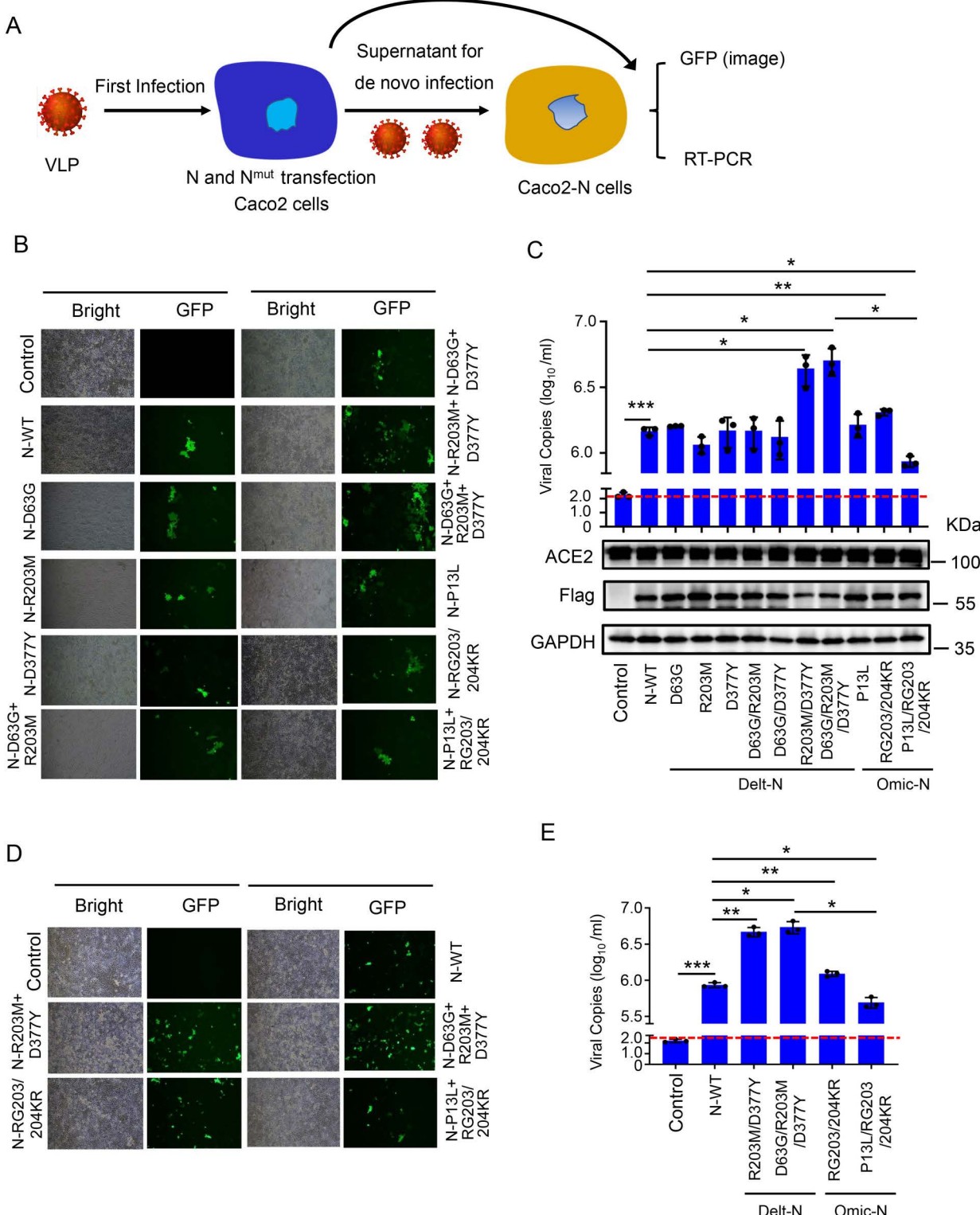

**Fig 2. The R203M and D377Y mutation of N protein promote SARS-CoV-2-VLP production in Caco-2 cells. (A)** Experimental scheme. Caco-2 cells were transfected with plasmids encoding different mutated N protein for 24 h, and then infected with SARS-CoV-2 GFP/ΔN (MOI = 0.5) for 2 h, washed and incubated for an additional 48 h. The cell culture medium was collected to infect the Caco-2-N cells for 48 h. GFP fluorescence was observed by microscopy and viral RNA in supernatant was determined by RT-qPCR assay. **(B, C)** Caco-2 cells were transfected with plasmids encoding different mutated N protein for 24 h, and then infected with SARS-CoV-2 GFP/ΔN (MOI = 0.5) for 2 h, washed and

incubated for an additional 48 h. GFP expression was observed in Caco-2 cells using microscopy at indicated time after inoculation (B). **(C)** The total RNA in supernatant of Caco-2 cells was extracted and RT-qPCR assays were conducted to determine viral copies. Cell lysates were analyzed by immunoblotting. **(D, E)** The above cell culture medium were collected to infect the Caco-2-N cells for 48 h. GFP expression was observed in Caco-2-N cells using microscopy after inoculation for 48 h (D). **(E)** The total RNA in supernatant of Caco-2-N cells was extracted and RT-qPCR assays were conducted to determine viral copies. Control means pcDNA3.1(+)-3 × flag empty plasmid (B–E). Data are representative of three independent experiments and one representative is shown. Error bars indicate SD of technical triplicates. Values are mean ± SEM. *P ≤ 0.05, **P ≤ 0.01, ***P ≤ 0.001, two-tailed Student's t-test.

the differences in replication dynamics among the three different variants may be associated with their ability to induce interferon. Next, the effect of mutated N proteins on interferon was explored. Mutated N proteins and luc-IFN-β plasmids were transfected in HEK293T cells which were then stimulated with interferon-specific activators Sendai virus (Sev) (Fig 4C), and Poly I:C [32] (S3A Fig), respectively. Results depicted that: original N protein inhibited IFN-β promoter level compared to control; mutations with both R203M and D377Y in Delta N protein inhibited IFN-β promoter level more strongly than original and Omicron N protein (Figs 4C and S3A). Next, mutant N protein plasmids were transfected in HEK293T-ACE2 cells (S3B and S3E Fig) or Caco-2 cells (Fig 4D and 4F) and then infected with trVLP. The qRT-PCR results exhibited that: original N protein suppressed mRNA levels of IFN-β compared to control, but slightly suppressed IFN-α and IFN-γ; among the variants, mutations in Delta variant having both R203M and D377Y suppressed mRNA levels of IFN-β most strongly (Figs 4D and S3B). The above results were also proved in caco-2 cells that were infected with original SARS-CoV-2 (Figs 4E and S3C and S3D). ELISA also demonstrated that the R203M/D377Y mutation mostly enhanced the inhibitory role of N protein in IFN-β induction (Figs 4F and S3E).

To further study whether the mutations in N protein affected viral replication through regulating IFN-β, we first transfected Caco-2 cells with different mutated N proteins, and then added Anifrolumab, a type I interferon receptor antagonist, which can block the activity of type I interferons [33]. Subsequently, the cells were infected with VLP (Fig 4G) and SARS-CoV-2 live virus (Fig 4H), respectively. The results showed that: Delta N protein having mutations containing sites R203M and D377Y promoted SARS-CoV-2 replication more strongly than the Omicron mutant N protein (containing sites P13L and RG203/204KR) and original N protein; the mutations P13L and RG203/204KR in the Omicron mutant N protein reduced its activity on promoting viral replication compared with original N protein. However, after pre-treated with Anifrolumab, the different mutated N proteins had no effect on the replication of VLP (Fig 4G) and SARS-CoV-2 live virus (Fig 4H). These results demonstrated that the mutations in N protein of different variants affected viral replication by regulating IFN.

### R203M/D377Y mutation enhancing N protein-caused suppression of RIG-I signaling

RNA viruses including SARS-CoV-2 were primarily recognized by intracellular RIG-I or MDA5 which in turn regulated the interferon pathway [34]. The previous study found that SARS-CoV-2 original N protein interacted with G3BP1 to prevent stress granule formation and to keep the cofactors G3BP1 and PACT from activating RIG-I [35] and SARS-CoV-2 infection activated the IFN response through MDA5-mediated sensing [36]. Therefore, we investigated whether these mutations of N protein alter N-G3BP1, N-PACT, N-RIG-I, or N-MAD5 interactions. CO-IP results showed that: the original and mutant N protein could interact with G3BP1, PACT and MDA5, and that the mutations in N proteins didn't alter the interaction with G3BP1(S4A Fig), PACT (S4B Fig) or MDA5

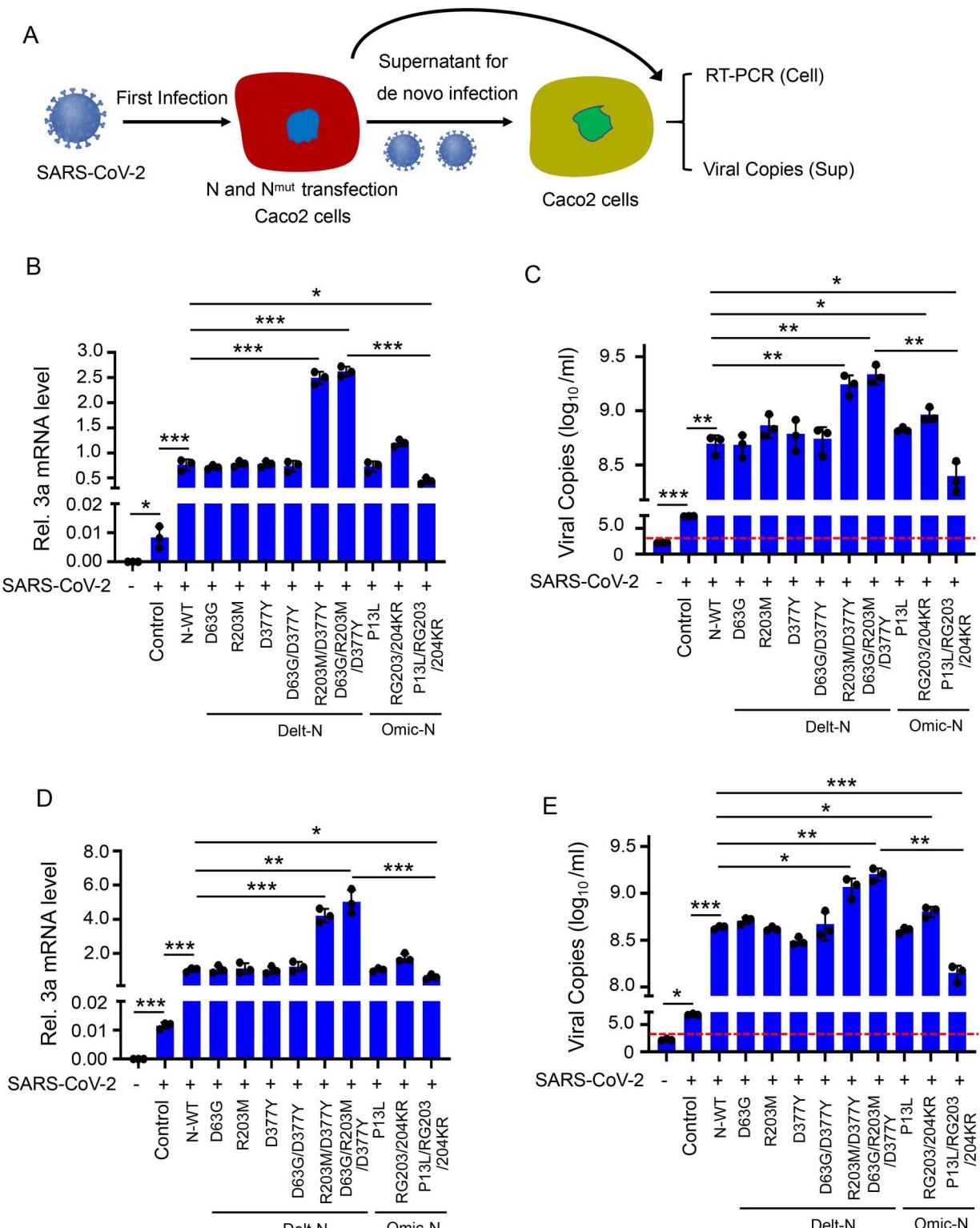

**Fig 3. The R203M and D377Y mutation of N protein promote SARS-CoV-2 production in Caco-2 cells. (A)** Experimental scheme. Caco-2 cells were firstly transfected with plasmids encoding different mutated N protein for 24 h, and then infected with SARS-CoV-2 (MOI = 0.5) for 2 h, washed virus and incubated fresh maintenance culture medium for an additional 24 h. Viral RNA in cell and viral copies in supernatant was determined by RT-qPCR assay. The cell culture medium include virus was collected to infect the untransfected Caco-2 cells for another 24 h. Viral RNA in cell and viral copies in supernatant was determined by RT-qPCR assay. **(B)** The total RNA in Caco-2 cells was extracted

and RT-qPCR assays were conducted to determine viral mRNA. **(C)** The total RNA in supernatant of Caco-2 cells was extracted and RT-qPCR assays were conducted to determine viral copies. **(D)** The total RNA in Caco-2 cells was extracted and RT-qPCR assays were conducted to determine viral mRNA. **(E)** The total RNA in supernatant of Caco-2 cells was extracted and RT-qPCR assays were conducted to determine viral copies. Control means pcDNA3.1(+)-3 × flag empty plasmid (B–E). Data are representative of three independent experiments and one representative is shown. Error bars indicate SD of technical triplicates. Values are mean ± SEM. *P ≤ 0.05, **P ≤ 0.01, ***P ≤ 0.001, two-tailed Student's t-test.

(S4C Fig) compared with original N protein. However, compared with original N protein, the R203M and D377Y mutation in Delta N protein inhibited its interaction with endogenous RIG-I protein, while Omicron mutant N protein displayed slight enhancement on the interaction of RIG-I both in HEK293T cells (S4D Fig) and Caco-2 cells (Fig 5A). In order to further exclude the influence of other intracellular proteins or RNA on the above experimental results, we used a prokaryotic expression system to express and purify RIG-I protein (labeled with GST tag, S4E Fig) and mutant N proteins (including Delta mutant N protein, Omicron mutant N protein, R203M and D377Y mutant N protein, RG203/204KR mutant N protein; all the protein was labeled with His tag, S4F Fig). The original purified His-tagged N protein was saved in our lab. The pull-down assay further confirmed the above experimental conclusion: compared with original N protein, the R203M and D377Y mutation in Delta N protein inhibited the direct interaction with RIG-I protein, while Omicron mutant N protein displayed enhancement on the interaction of RIG-I (Fig 5B). The dual luciferase assay for IFN-β further confirmed that the N protein mutations inhibited the production of interferon through RIG-I, rather than MDA5 (S4G Fig). Immuno-fluorescence was further to demonstrate the co-location between the RIG-I and mutant N proteins in SARS-CoV-2-VLP infected Caco2 cells. Results revealed that: original N, RIG-I and GFP-tagged VLP were co-located in the cytoplasm. Compared with original N, the levels of colocalization among Delta having both N proteins R203M and D377Y with RIG-I and VLP were reduced. Compared with Delta mutant N, Omicron mutant N protein promoted the colocalization of RIG-I and VLP (S5 Fig). The activated RIG-I protein caused antiviral effects by recruiting downstream MAVS proteins [37]. Therefore, it was imperative to demonstrate whether mutated N proteins affected the binding of RIG -I and MAVS. Results of this study proved that: Delta having both N protein mutations R203M and D377Y inhibited the interaction of RIG-I and MAVS proteins compared to original N protein, while Omicron mutant N protein slightly altered the interaction of RIG-I and MAVS; Omicron N protein promoted interaction between RIG-I and MAVS proteins compared to Delta N protein (Fig 5C). Previous studies had demonstrated that N protein interacts with TRIM25 which reduces the ubiquitination of RIG-I [38–40]. Next, we further investigated the impact of the mutated N protein on the RIG-I-TRIM25 interactions and its ubquitination. The CO-IP results proved that the mutated N protein does not affect the interaction between RIG-I and TRIM25 (S6A Fig), nor does it influence the ubiquitination of RIG-I by TRIM25 (S6B Fig). Next, the phosphorylation levels of TBK1 and IRF3 downstream of MAVS were examined both in HEK293T (Fig 5D) and Caco-2 cells (Fig 5F). Results revealed that: mutation having both N protein R203M and D377Y in Delta strain inhibited phosphorylation levels of TBK1 and IRF3, and suppressed IRF3 entry into nucleus compared to original N protein (Fig 5E), while Omicron mutant N protein had no such effects; Omicron N protein promoted phosphorylation levels of TBK1 and IRF3, and IRF3 entry into nucleus compared to Delta N protein (Fig 5D–5F). These results demonstrated that N protein of Delta variant inhibited the interaction between RIG-I and MAVS compared to original strain, thus suppressing the phosphorylation levels of TBK1 and IRF3.

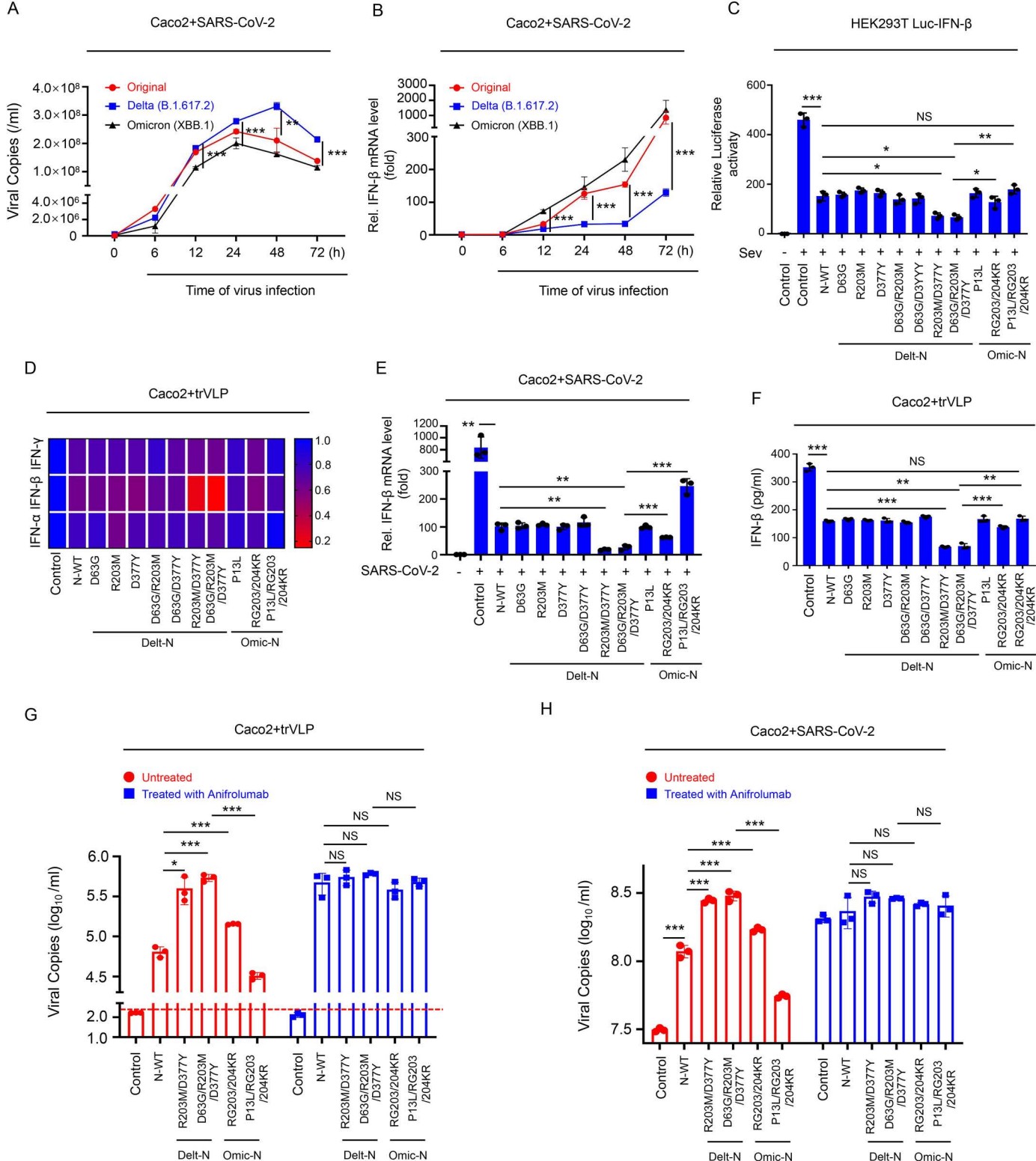

**Fig 4. The R203M and D377Y mutation of N protein inhibited IFN-β production.(A, B)** Caco-2 cells were respectively infected with original strain, Delta strain or Omicron strain (MOI = 0.5) at indicated time (6, 12, 24, 48, 72 hours), viral copies in supernatant were determined by RT-qPCR assay (A) and the total RNA in cells was extracted and RT-qPCR assays were conducted to determine IFN-β (B). **(C)** HEK293T cells were transfected with IFN-β luciferase reporter pIFN-β-Luc, pPRL-TK and different mutated N protein for 24 h and then infected with SeV (MOI = 0.1) for 16 h. Cell lysates were harvested, IFN-β-Luc reporter activity was determined by dual luciferase reporter assays. **(D, F)** Caco-2 cells were transfected with plasmids encoding different mutated N protein for 24 h, and

then infected with SARS-CoV-2 GFP/ΔN (MOI = 0.5) for 2 h, washed and incubated for an additional 48 h. The total RNA in cells was extracted and RT-qPCR assays were conducted to determine IFN-α, IFN-β and IFN-γ (D). Supernatants were analyzed by ELISA for IFN-β (F). **(E)** Caco-2 cells were transfected with plasmids encoding different mutated N protein for 24 h, and then infected with SARS-CoV-2 (MOI = 0.5) for 2 h, washed and incubated for an additional 48 h. The total RNA in cells was extracted and RT-qPCR assays were conducted to determine IFN-β. **(G)** Caco-2 cells were transfected with plasmids encoding different mutated N protein for 24 h, then treated with Anifrolumab (2 μg/ml) and infected with SARS-CoV-2 GFP/ΔN (MOI = 0.5) for 2 h, washed and incubated with Anifrolumab (2 μg/ml) for an additional 48 h. viral copies in supernatant was determined by RT-qPCR assay. **(H)** Caco-2 cells were transfected with plasmids encoding different mutated N protein for 24 h, then treated with Anifrolumab (2 μg/ml) and infected with SARS-CoV-2 (MOI = 0.5) for 2 h, washed and incubated with Anifrolumab (2 μg/ml) for an additional 24 h. viral copies in supernatant was determined by RT-qPCR assay. Control means pcDNA3.1(+)-3 × flag empty plasmid (C–H). Data are representative of three independent experiments and one representative is shown. Error bars indicate SD of technical triplicates. Values are mean ± SEM. *P ≤ 0.05, **P ≤ 0.01, ***P ≤ 0.001, NS mean no significant difference, two-tailed Student's t-test.

## R203M/D377Y mutation promoting the RNA-binding activity of N protein and attenuating the interaction of RIG-I and viral RNA

N protein as the structural protein of SARS-CoV-2 has an important function to bind viral RNA during viral packaging process [41]. Therefore, mutant N proteins ability to bind viral RNA was further investigated. Mutant N protein plasmids were transfected in HEK293T cells, and co-incubated the cell lysates with *in vitro* purified SARS-CoV-2 RNA (S7A Fig). RNA binding protein immunoprecipitation (RIP) experiments depicted that: original N protein could bind viral RNA compared to control group; Delta variant with N protein R203M and D377Y increased the ability to bind viral RNA compared to original N protein; Omicron N protein inhibited the capacity to bind viral RNA compared to Delta N-protein (Fig 6A). Next, Caco-2 cells were transfected with mutant N protein plasmids and then infected with trVLP (S7B Fig). Like above results, namely: original N protein could bind viral RNA in large amounts compared to control; Delta mutant containing both N proteins R203M and D377Y increased the ability to bind viral RNA compared to original N protein; Omicron N protein inhibited the binding ability of viral RNA compared to Delta N protein (Fig 6B). Mutation of N protein RG203/204KR improved the binding ability to viral RNA compared to original N protein (Fig 6A and 6B). This was consistent with previous findings that mutation of N protein RG203/204KR promoted SARS-CoV-2 replication [28,29]. It was proved that N proteins could interact with RIG-I, as RIG-I could recognize viral RNA [37]. Next, it was investigated whether mutant N proteins competed with RIG-I to bind viral RNA and inhibited RIG-I-regulated interferon pathway. Mutant N proteins and RIG-I plasmids were transfected in Caco-2 cells, and co-incubated the cell lysates with *in vitro* purified viral RNA using RIG-I as IP (S7C Fig). RIP experiments exhibited that original N protein inhibited the binding ability of RIG-I and viral RNA compared to control; Delta variant containing both N protein R203M and D377Y inhibited binding ability of RIG-I and viral RNA compared to original N protein, while Omicron N protein slightly promoted binding of RIG-I and viral RNA; Omicron N protein promoted binding of RIG-I and viral RNA compared to Delta N protein (Fig 6C). Finally, Caco-2 cells were transfected with mutant N protein and RIG-I plasmids and then infected with trVLP (S7D Fig). Results were in accordance with above conclusions (Fig 6D). The mutation of N protein RG203/204KR inhibited binding ability of RIG-I and viral RNA compared to original N protein (Fig 6C and 6D). The above conclusion was further confirmed through the Pull-down experiment (incubation with the same quantity of RIG-I protein and viral RNA, detection of the level of RIG-I binding viral RNA in the presence of different mutant N proteins). The results showed that: compared with control, original N protein could inhibit the binding of RIG-I to viral RNA; compared with original N protein, the Delta variant containing both N protein R203M and D377Y could significantly inhibit the binding of RIG-I to viral RNA, while Omicron N protein slightly promoted binding of RIG-I and viral RNA; compared to Delta N protein, Omicron N protein promoted RIG-I binding to viral RNA.

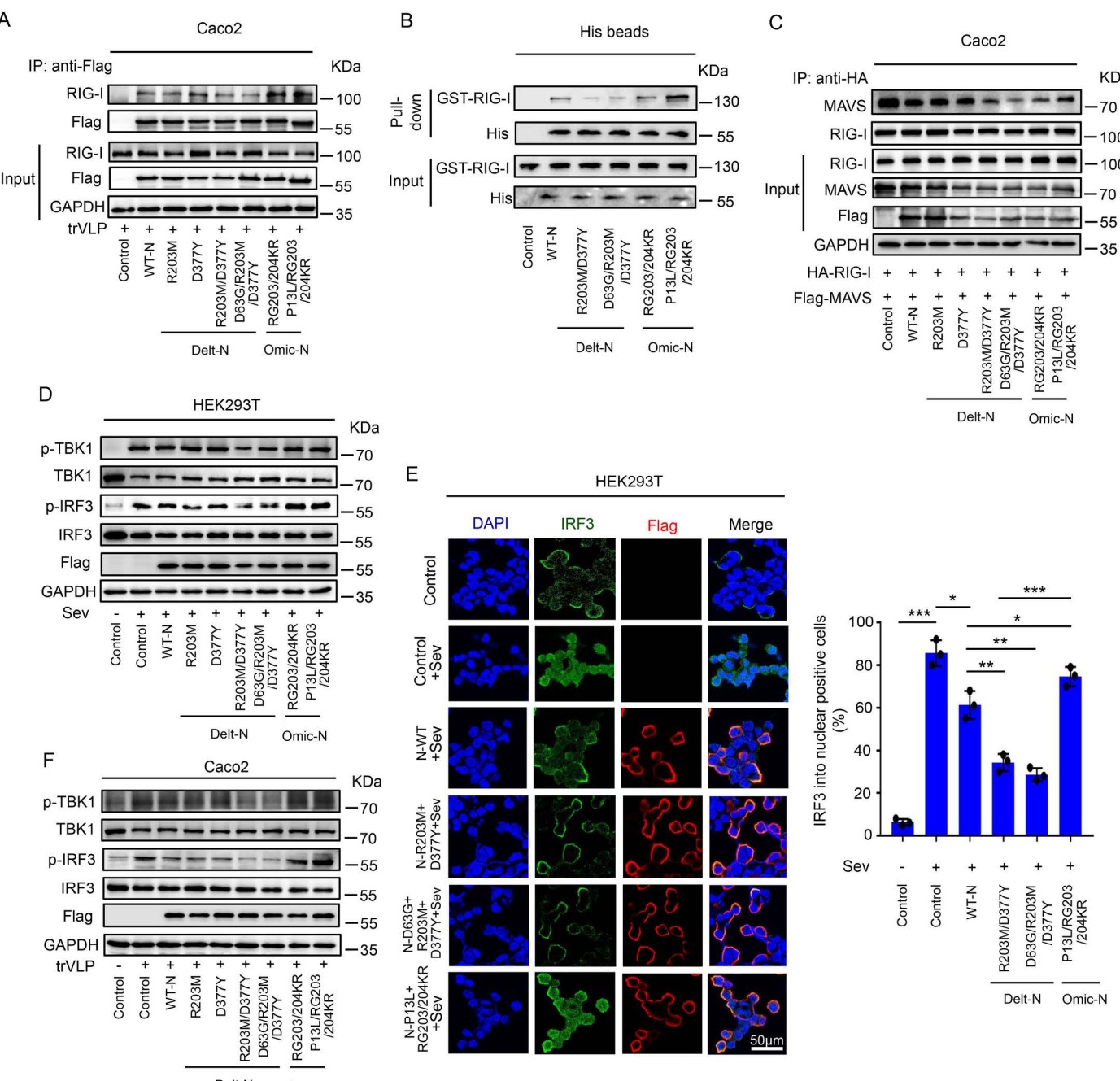

**Fig 5. The R203M and D377Y mutation of N protein inhibited IFN-β production through regulating RIG-I-MAVS-TBK1-IRF3 pathway.** **(A)** Caco-2 cells were transfected with plasmids encoding different mutated N protein for 24h and then infected with SARS-CoV-2 GFP/ΔN (MOI = 0.5) for 2h, washed and incubated for an additional 48h. Cell lysates were immunoprecipitated using anti-Flag antibody, and analyzed using anti-Flag, anti-RIG-I and anti-GAPDH antibody. Cell lysates (40 μg) was used as Input. **(B)** Purified RIG-I-GST protein (2 μg) and purified mutant N-His protein were incubated for 60 min, part of the mixture was used as Input, then the mixture was pulldown using His Beads. Pulldown analysis of purified RIG-I and N protein. **(C)** Caco-2 cells were co-transfected with plasmids encoding different mutated N protein with RIG-I and MAVS for 24h. Cell lysates were immunoprecipitated using anti-HA antibody, and analyzed using anti-Flag, anti-RIG-I, anti-MAVS and anti-GAPDH antibody. Cell lysates (40 μg) was used as Input. **(D)** HEK293T cells were transfected with plasmids encoding different mutated N protein for 24h and then infected with SeV (MOI = 0.1) for 16h. Cell lysates were analyzed by immunoblotting. **(E)** HEK293T cells were transfected with plasmids encoding different mutated N protein for 24h and then infected with SeV (MOI = 0.1) for 16h. Nucleus marker DAPI (blue), IRF3 (green) and Flag- mutated N protein (red) were then visualized with confocal microscopy. Statistics of positive cells entering the nucleus with IRF3. **(F)** Caco-2 cells were transfected with plasmids encoding different mutated N protein for 24h, and then infected with SARS-CoV-2 GFP/ΔN (MOI = 0.5) for 2h, washed and incubated for an additional 48h. Cell lysates were analyzed by immunoblotting. Control means pcDNA3.1(+)-3 × flag empty plasmid **(A, C–F)** or purified His-empty protein (B). Data are representative of three independent experiments and one representative is shown. Error bars indicate SD of technical triplicates. Values are mean ± SEM. *P ≤ 0.05, **P ≤ 0.01, ***P ≤ 0.001, two-tailed Student's t-test.

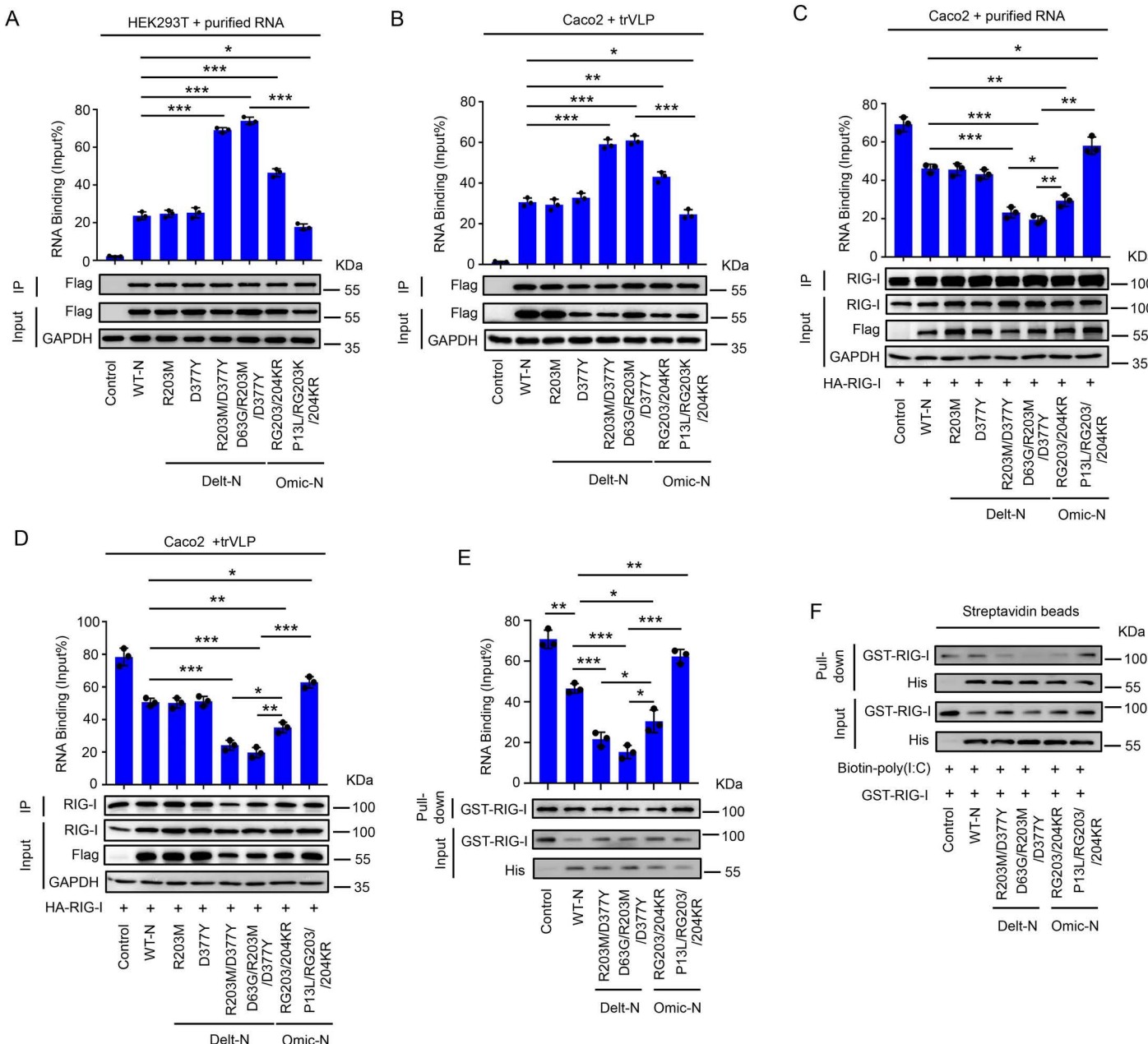

**Fig 6. The R203M and D377Y mutation of N protein promote the activity of RNA-binding. (A)** HEK293T cells were transfected with plasmids encoding different mutated N protein for 24 h, Cell lysates were collected and incubated with purified RNA of SARS-CoV-2 GFP/ΔN VLP (10 μg) for 60 min, the level of VLP RNA retrieval (% input) with mutant and wide-type (WT) N protein were analyzed by RT-qPCR assay using specific viral N gene (ORF3a), Cell lysates were analyzed by immunoblotting. **(B)** Caco-2 cells were transfected with plasmids encoding different mutated N protein for 24 h and then infected with SARS-CoV-2 GFP/ΔN (MOI = 0.5) for 2 h, washed and incubated for an additional 48 h, the level of viral RNA retrieval (% input) with mutant and WT-N protein were analyzed by RT-qPCR assay using specific ORF3a gene, Cell lysates were analyzed by immunoblotting. **(C)** Caco-2 cells were transfected with plasmids encoding different mutated N protein plus RIG-I protein for 24 h, Cell lysates were collected and incubated with purified RNA of SARS-CoV-2 GFP/ΔN VLP (10 μg) for 60 min, the level of VLP RNA retrieval (% input) with RIG-I were analyzed by RT-qPCR assay using specific viral N gene (ORF3a), Cell lysates were analyzed by immunoblotting. **(D)** Caco-2 cells were transfected with plasmids encoding different mutated N protein plus RIG-I protein for 24 h and then infected with SARS-CoV-2 GFP/ΔN (MOI = 0.5) for 2 h, washed and incubated for an additional 48 h. The level of viral RNA retrieval (% input) with RIG-I protein were analyzed by RT-qPCR assay using specific ORF3a gene, Cell lysates were analyzed by immunoblotting. **(E)** Purified RIG-I-GST protein (2 μg), purified mutant N-His protein (2 μg) and purified RNA of SARS-CoV-2 GFP/ΔN VLP (10 μg) were incubated for 60 min, then the mixture were pulldown using GST Magnetic Agarose Beads, the level of VLP RNA retrieval (% input) with RIG-I were analyzed by RT-qPCR assay using specific viral N gene (ORF3a), Cell lysates were analyzed by immunoblotting. **(F)** Purified RIG-I-GST protein (2 μg), purified mutant N-His protein (2 μg) and Biotin-tagged poly (I:C) (2 μg) were incubated for 60 min, then the mixture was pulldown using Streptavidin Beads. Pulldown analysis of purified RIG-I binding to poly (I:C) with or without N protein. Control means pcDNA3.1(+)-3 × flag empty plasmid (A-D) or purified His-tagged empty protein (E, F). Data are representative of three independent experiments and one representative is shown. Error bars indicate SD of technical triplicates. Values are mean ± SEM. *P ≤ 0.05, **P ≤ 0.01, ***P ≤ 0.001, two-tailed Student's t-test.

Mutation of N protein RG203/204KR inhibited the RIG-I binding to viral RNA compared to original N protein (Fig 6E). Furthermore, biotin streptavidin affinity experiments proved that: compared to original N protein, the Delta variant containing both N protein R203M and D377Y significantly inhibited biotin labeled poly I:C binding to RIG-I; compared with Delta N protein, Omicron N protein promoted biotin labeled poly I:C binding to RIG-I (Fig 6F). Taken together, the above results reflected that N protein of Delta variant inhibited the binding of RIG-I to viral RNA by binding to own viral RNA, thus inhibiting the RIG-I activation.

## R203M/D377Y mutation promotes virus replication and induces mice lung injury via inhibiting the IFN-β production

The biological role of mutant N proteins in SARS-CoV-2 in enhancing immunosuppression was then evaluated. We explored the effect of mutant N proteins using an AAV-lung-N C57BL/6 mouse model that previously used in our laboratory. C57BL/6 mice were subjected to tail vein injection with AAV-Lung-control, AAV-Lung-original-N, AAV-Lung-R203M/D377Y-N, AAV-Lung-Delta (D63G/R203M/D377Y)-N, AAV-Lung-Omicron (P13L/RG203/204KR)-N. After four weeks, the above mice were infected with Vesicular Stomatitis Virus (VSV) which can activate the RIG-I-MAVS signaling pathway to regulate interferon production. After 16 hours of viral infection, blood was collected to detect the mRNA levels of IFN. The results showed that: VSV infection significantly promoted the expression of IFN-β compared to the no infected group. Compared with the control group, the original N group significantly inhibited the expression of IFN-β. Compared with the original N group, Delta variant with N protein R203M and D377Y significantly inhibited the expression of IFN-β. Compared with the Delta variant N group, the Omicron variant N group significantly promoted the expression of IFN-β (Fig 7A left). The protein level of IFN-β in mice serum was similar with above results (Fig 7B). Although VSV infection can activate the mRNA levels of IFN-α (Fig 7A middle) and IFN-γ (Fig 7A right), there is no significant difference between the different groups mentioned above, indicating that the mutated N protein specifically affects the production of IFN-β. After 48 hours of VSV infection, euthanize the mice and take their lungs to detect the mRNA levels of IFN and VSV. The results showed that: VSV infection significantly promoted the expression of IFN-α, IFN-β and IFN-γ compared to the no infected group in the lungs of mice. Compared with the control group, the original N group significantly inhibited the expression of IFN-β. Compared with the original N group, Delta variant with N protein R203M and D377Y significantly inhibited the expression of IFN-β. Compared with the Delta variant N group, the Omicron variant N group significantly promoted the expression of IFN-β (Fig 7C left). However, the mRNA levels of IFN-α (Fig 7C middle) and IFN-γ (Fig 7C right) had no significant difference between the different groups. We also tested the replication of VSV in mouse lung tissue at the same time. The results showed that: compared with the no infected groups, VSV successfully replicated in mouse lungs. Compared with the control group, the original N group significantly promoted VSV replication. Compared with the original N group, Delta variant with N protein R203M and D377Y significantly promoted VSV replication. Compared with the Delta mutant N group, the Omicron mutant N group significantly inhibited VSV replication (Fig 7D). Notably, Hematoxylin & eosin staining analyses indicated that: compared with the no infected group, VSV infection significantly promoted the inflammatory lesions (red arrow) and tissue injuries (red circle) of mouse lung, compared with the control group, the original N group slightly promoted inflammatory lesions (red arrow) and tissue injuries (red circle) of mouse lung. Compared with the original N group, Delta variant with N protein R203M and D377Y significantly promoted inflammatory lesions (red arrow) and tissue injuries (red circle) of mouse lung. Compared with the

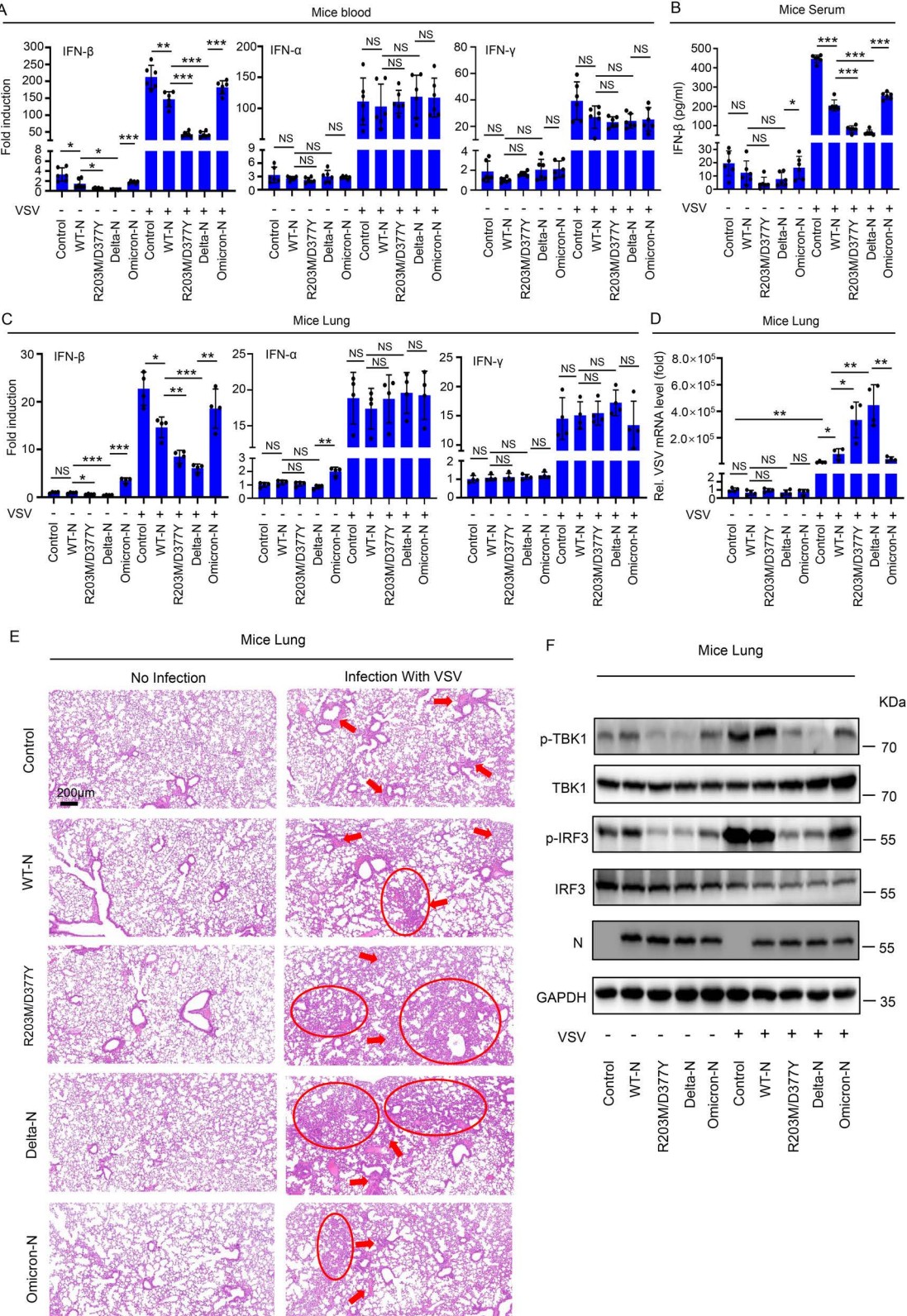

**Fig 7. R203M/D377Y mutation promotes virus replication and induces mice lung injury via inhibiting the IFN-β production.** C57BL/6 genetic background mice were tail vein injection with 300 μl containing $5 \times 10^{11}$ vg of AAV-Lung-EGFP (n = 12) or AAV-Lung-N or mutant N (included R203M/D377Y, Delta-N and Omicron, n = 12 respectively), after fourth

weeks, the above mice were infected with VSV ($4 \times 10^8$ PFU per mouse) by intravenous injection (n = 6). **(A, B)** After 16 hours infection, blood was collected for each group from the orbit. The total RNA in blood was extracted and RT-qPCR assays were conducted to determine IFN-α, IFN-β and IFN-γ (A). Serum were analyzed by ELISA for IFN-β (B). Points represent the value of each serum samples. **(C–F)** After 48 hours infection, mice were euthanasia and lung tissue were collected. The total RNA in lung was extracted and RT-qPCR assays were conducted to determine IFN-α, IFN-β and IFN-γ (C), the VSV mRNA level in the lung was also detected (D). Points represent the value of each serum samples. Histopathology analysis of the mice's lungs (E). Scale bar, 200 μm. Red arrow indicated inflammatory lesions and red circle indicated tissue injuries. The lysates of Lung tissue were analyzed by immunoblotting (F). Control means AAV-Lung-EGFP infected mice **(A–F)**. Data are representative of three independent experiments and one representative is shown. Error bars indicate SD of technical triplicates. Values are mean ± SEM. *P ≤ 0.05, **P ≤ 0.01, ***P ≤ 0.001, NS mean no significant difference, two-tailed Student's t-test.

Delta mutant N group, the Omicron mutant N group significantly inhibited inflammatory lesions (red arrow) and tissue injuries (red circle) of mouse lung (Fig 7E). Finally, the phosphorylation levels of TBK1 and IRF3 downstream of MAVS were examined in mouse lung. Results revealed that: mutation having both N protein R203M and D377Y in Delta strain inhibited phosphorylation levels of TBK1 and IRF3 compared to original N protein, Omicron N protein promoted phosphorylation levels of TBK1 and IRF3 compared to Delta N protein (Fig 7F). The above experimental results demonstrate that mutations at the R203M and D377Y sites of the N protein can inhibit the production of IFN-β and increase its immunosuppressive ability by regulating the RIG-I-MAVS signaling pathway at the mouse level.

Taken together, the above results demonstrated that R203M and D377Y of N protein in Delta strains exert different activities on binding viral RNA and regulating RIG-I-mediated antiviral response (Fig 8).

## Discussion

As a structural protein of SARS-CoV-2, the nucleocapsid protein (N) plays a crucial role in the virus's life cycle [22]. Previous studies have found that mutations at N protein R203K/G204R sites promoted own replication [28,29]. In the present study, mutations at the R203M and D377Y sites of the N protein were observed to enhance self-replication. These two sites were found in Delta variant but not in Omicron and its subvariants. The molecular mechanism was that: mutations in N protein R203M and D377Y sites specifically inhibited IFN-β production by inhibiting the interaction between RIG-I and MAVS. This inhibition subsequently prevents the phosphorylation of downstream TBK1 and IRF3. Additionally, these mutations inhibited binding of RIG-I and viral RNA by promoting binding to own RNA, thus inhibiting RIG-I activation and promoting viral replication.

In comparison to the S protein, the SARS-CoV-2 N protein has shown a high degree of conservation in its amino acid sequence. However, some mutations in N protein had been observed with the emergence of variants [29]. This study identified three major mutations in the N protein of the Delta variant (D64G, R203M, and D377Y) and three major mutations in the Omicron variant (P13L, R203K, and G204R) relative to the original strain. Mutations at N protein sites R203K and G204R in Omicron were also present in previous Alpha and Gamma variants, but not in Delta. Branch model was employed to demonstrate that Delta's N protein was less constrained compared to other SARS-COV-2 variants, suggesting the possibility of positive selection.

Next, the effect of N proteins site mutation on viral replication was demonstrated using SARS-CoV-2-GFP/ΔN-trVLP system [31] and further explored with live SARS-CoV-2 in P3 laboratory. It was found that Delta mutant N protein promoted viral replication compared to original N protein, while Omicron mutant N protein reduced the promotion of viral replication compared to Delta mutant N protein. Previous studies found that Delta variant had

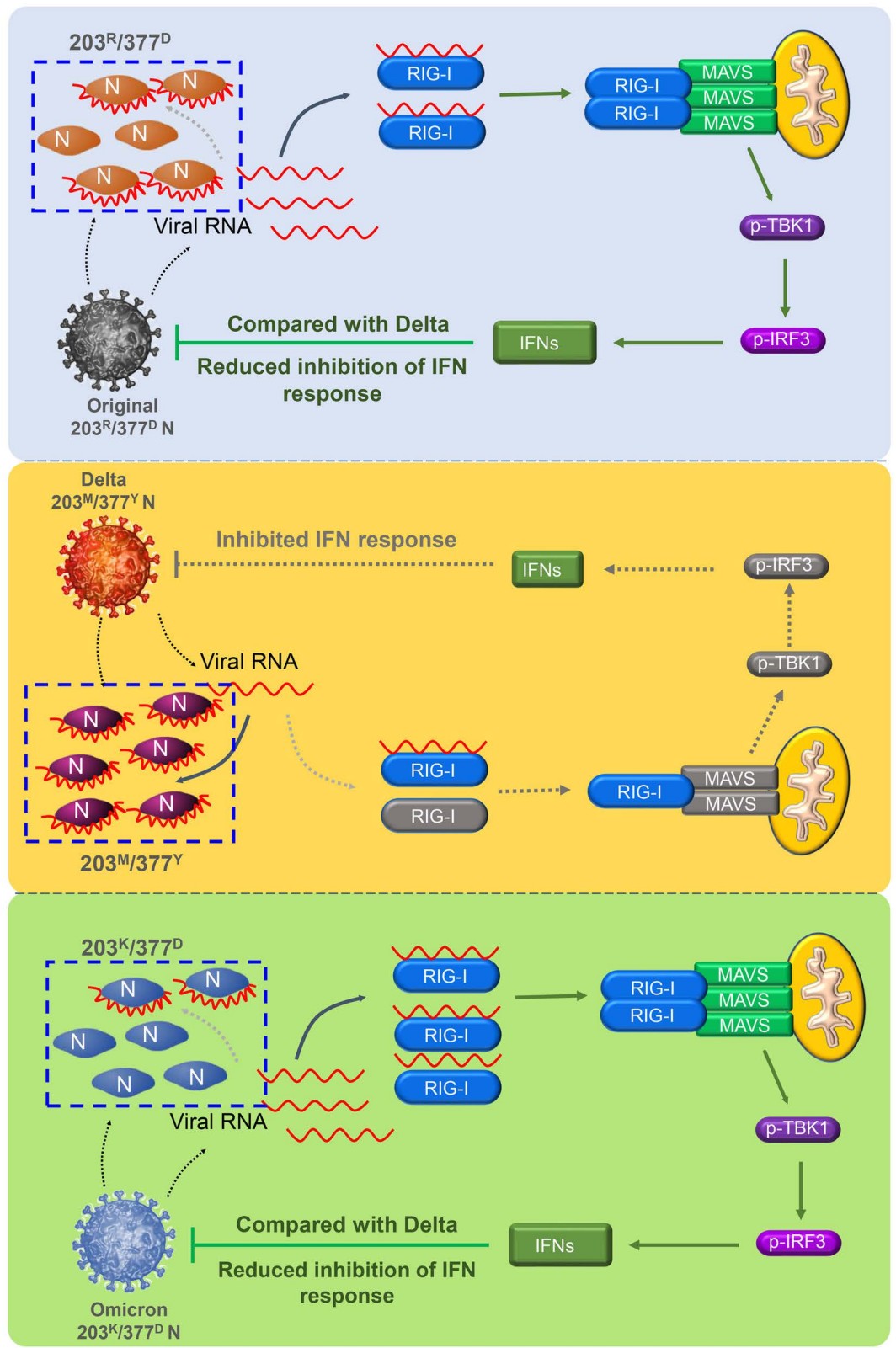

**Fig 8. The model of this article.** The R203M and D377Y of N protein in Delta strains exert different activities on binding viral RNA and regulating RIG-I-mediated antiviral response.

higher potential to infect and replicate in cellular models and hosts compared to the original strain [9,10]. Despite having a reduced replication ability, the Omicron variant exhibits a higher capacity for spread and immune escape [11,12]. These findings elucidate the impact of N protein mutations on the replication of Delta and Omicron variants. However, further research is necessary to investigate the effects on transmission and immune escape abilities of various variants.

The innate immune system had antiviral role and acted as body's first defense line. RIG-I-MAVS-TBK1-IRF3 signaling pathway resists RNA virus infection [42]. We previously found that original N protein of SARS-CoV-2 inhibited IFN-β production through interacting with RIG-I [24]. In the present study, it was found that Delta variant N protein inhibited the interaction between RIG-I and downstream MAVS, inhibited the phosphorylation of TBK1 and IRF3, and inhibited IFN-β production; whereas N protein of Omicron had little effect on IFN-β. More interesting, We found that the mutant N protein inhibited the production of IFN-β, but has no effect on the production of IFN-α. We speculated that this may be due to two reasons. First, in the cells we used, the expression of IFN-β is higher than that of IFN-α, making it more susceptible to changes and detection. Second, there are multiple homologous genes for IFN-α in the genome, and our primers may not be able to accurately assess the overall expression of IFN-α. Previous studies revealed that mutations in N protein R203K/G204R promoted own replication by facilitating its binding to viral RNA [28]. We found that Delta N protein promoted its binding to viral RNA compared with original N protein, and similar to previous report, mutation at N protein R203K/G204R site also promoted its binding to viral RNA. The N protein of Omicron inhibited its binding to viral RNA to certain extent, which might explain molecular mechanism of Omicron variant's low replication in the host. The RIP experiments demonstrated that the N protein of the Delta variant inhibits the binding of RIG-I to viral RNA. This finding elucidates the molecular mechanism by which the N protein suppresses innate immunity by preventing RIG-I from binding to viral RNA. Seven Delta variant N protein clones were constructed by permutation for determining specific action site of Delta variant N protein, and it was proved that Delta variant N protein had mutations at sites R203M and D377Y with aforementioned biological functions. These two mutation sites were also unique to Delta which explained the relatively higher replication ability of Delta variant. Notably, the phenotypic effects of these mutations were observed using the trVLP system. Ideally, these findings should be confirmed using actual infectious SARS-CoV-2 clones harboring the same mutations, but technically it is difficult to make infectious clones with specific mutations in the nucleocapsid. The D377Y and R203M mutations are located in the C-terminal dimerization domain and the linkage region, respectively, rather than the N-terminal RNA binding domain (N-NTD). Although experimental evidences that showed the N-NTD functioned as classical RNA binding region, the C-terminal dimerization domain was also proved to be involved in capturing the RNA genome and the linkage region anchored the ribonucleoprotein complex to the viral membrane [43–45]. Thus, our conclusion that the D377Y and R203M mutations in N protein enhanced its activity of capturing viral RNA is understandable. We did not know whether the binding sequence of viral RNA to N protein was the same as that to RIG-I protein. However, our result indicated when the N protein-viral RNA complex forms, the recognition and binding of viral RNA by the RIG-I protein were attenuated. It could be speculated that N protein-viral RNA complex formed a kind of dense RNA folding which could not bind to RIG-I.

In summary, this study found that mutations at N protein sites R203M and D377Y in SARS-CoV-2 promoted self-replication. This suggested that in designing SARS-CoV-2 vaccines and antibodies, attention must also be given to other proteins, especially N protein mutation in addition to the mutation of S protein.

## Materials and methods

### Ethics statement

All animal studies were performed in accordance with the principles described by the Animal Welfare Act and the National Institutes of Health Guidelines for the care and use of laboratory animals in biomedical research. The experimental protocols and all procedures involving mice were approved by the Institute of Laboratory Animal Science, Jinan University (approval number: 20220302-25). All the mice were sacrificed by euthanasia.

### Animal studies

Wild-type C57BL/6 mice were purchased from Guangdong Medical Laboratory Animal Center, Guangzhou, China. Mice were cultivated and maintained under specific pathogen-free conditions at Jinan University. Four weeks old mice were infected with 300 μl of AAV-Lung-EGFP-original-N/ R203M/D377Y-N/ Delta (D63G/R203M/D377Y)/ Omicron (P13L/RG203/204KR)-N ($5 \times 10^{11}$ vg) or AAV-Lung-EGFP (purchased from OBiO Technology, Shanghai, China) by the tail vein injection. After four weeks, the mice were infected with the VSV ($4 \times 10^8$ PFU per mouse) by intravenous injection. After 16 hours infection, blood was collected for each group from the orbit. After 48 hours infection, the mice were sacrificed, and their lung tissues were collected for histopathology analysis.

### The system of SARS-CoV-2 virus-like particle (trVLP)

The full-length SARS-CoV-2 GFP/ΔN cDNA was kindly provided by Dr. Qiang Ding, Tsinghua University. Then, the viral RNA transcript transcribed in vitro was electroporation into Caco-2 cells stably expressing SARS-Cov-2 N gene (i.e., Caco-2-N cells). Within 48 hours, GFP fluorescence was observed, and after 96 hours, cytopathic effect (CPEs) was observed, indicating the production and reproduction of recombinant SARS-CoV-2 virus like particles (trVLP). Cell culture supernatants were obtained and centrifuged at 1000 rcf for 10 minutes to remove cell fragments. Then, using 0.22 μ M filter membrane filter them. Divide all viruses into tubes and freeze at −80°C. Experiments related to authentic SARS-CoV-2, include Original strain (accession number is MN908947.3), Delta strain (B.1.617.2, accession number is OX014251.1) and Omicron (XBB.1, accession number is PP848079.1) were conducted in Guangzhou Customs District Technology Center ABSL-3 Laboratory.

### Cell culture

human embryonic kidney cell lines (HEK-293T) and Human colorectal adenocarcinoma cell lines (Caco-2) were purchased from the American Type Culture Collection (ATCC). The above cells were cultured in DMEM medium that required supplementation with 10% fetal bovine serum, 100 U/ml penicillin, and 100% μg/ml streptomycin. The cells were cultured at 37°C with 5% $CO_2$. The above cells are passaged every 24 hours.

### Reagents and antibodies

Lipofectamine 2000 (11668019) were purchased from Sigma-Aldrich, Invitrogen, Trizol reagent (15596018) were purchased from Ambion company. RNase inhibitor (N2515) purchased from Promrga. Dual Luciferase Reporter Gene Assay Kit (N2515) was purchased from Beyotime.

Anti-SARS-CoV-2-N (A20021) and ACE2 (A4612) were purchased from ABclonal Technology. Anti-Flag (F3165), HA (H6908) and GAPDH (G8759) antibodies were purchased from Sigma. Anti-RIG-I (D14G6), anti-MAVS (D5A9E), Anti-TBK1 (E8I3G), anti-p-TBK1

(D52C2), Anti-IRF3 (D6I4C), and anti-p-IRF3 (E7J8G) antibodies were purchased from Cell Signaling Technology. The immunoglobulin G (IgG) of control mice and rabbit used in the CO-IP experiment was purchased from Invitrogen, while the IgG of Dylight 649, cy3, and FITC labeled mice and rabbits in the immunofluorescence secondary antibody were purchased from Abbkine.

### RNA extract and qRT-PCR

According to the manufacturer's instructions, using Trizol reagent (Invitrogen, Carlsbad, CA) to extract total RNA. We used Roche LC480 for relative quantitative analysis of target genes by Real-time quantitative reverse transcriptase-PCR (qRT-PCR). To quantify viral copies in cell supernatant, ten-fold serial dilutions of plasmid containing viral protein ($1 \times 10^1$ to $1 \times 10^8$ template copies per reaction) were analyzed by qRT-PCR, and total viral copies were calculated from Ct values using the resulting standard curves. We used Primer-blast, NCBI ([www.ncbi.nlm.nih.gov](www.ncbi.nlm.nih.gov)) to design Real-time PCR primers and list the sequences in S1 Table.

### Clone construction

Plasmids pcDNA3.1(+)-3 × flag-WT-N/MAVS, plasmids pCAGG-HA-RIG-I were stored in our own laboratory. N-specific site mutation plasmid of Delta variant (D63G, R203M, D377Y, D63G plus R203M, D63G plus D377Y, R203M plus D33Y and D63G plus R203M plus D377Y) or Omicron variant (P13L, RG203/204KR and P13L plus RG203/204KR) is constructed by using the WT-N plasmid as the template and Quick Mutation Site-Directed Mutagenesis Kit (No. D0206M, Beyotime) according to the manufacturer's instructions. The sequences of PCR primers were listed in S2 Table

### Stable cell lines construction

In this work, we had successfully constructed stably expressed N-lentivirus and ACE-2-lentivirus cell lines. The procedure was as follows: Firstly, the plasmid pLenti-3 × flag-N/ACE-2 or control empty vector were co-transfected into HEK293T cells using Lipo2000 with packaging plasmids psPAX2 and pMD2G. Secondly, replace the old medium with fresh medium containing 2% FBS 12 hours after transfection. Then, 36 hours and 60 hours after transfection, collected the cell supernatants containing lentivirus, and used 0.45 μm filter filtration. The cells were infected with the collected lentivirus for 24 hours by adding polystyrene (Sigma, TR-1003). Finally, positive cells were screened using puromycin (Sigma, P8833) for another 4–7 days.

### Western-blot

The cells were collected at indicated time and washed twice with cold PBS, then lysed in lysis buffer. Protein concentration was detected with the Bradford assay (Bio-Rad, Richmond, CA). Cell lysates (50 μg) were electrophoresed on 8–12% SDS PAGE and transferred to nitrocellulose membranes (Amersham, Piscataway, NJ). Nonspecific bands of NC membranes were blocked using 5% skim milk for 2 hours. NC membranes were washed three times with PBS containing 0.1% Tween (PBST) and incubated with indicated antibodies. Protein bands were visualized using a Bio-Rad Image Analyzer (Serial No. 733BR3722).

### Co-immunoprecipitation assay

Co-transfected HEK293T and Caco-2 cells with indicated plasmids for 24 hours. Collected and lysed the above cells. Rotated the lysates at 4°C for 30 minutes and then centrifuged at 13000 g for 10 minutes to remove excess cell fragments. Collected cell supernatants, take out a

portion of the supernatants as input, incubated the remaining lysates with designated antibodies at 4°C overnight, then mix with protein G Sepharose beads (GE Healthcare, Milwaukee, WI, USA) at 4°C for 2 hours. The immunoprecipitates were washed 4–6 times with eluent, added with protein loading buffer, boiled in boiling water for 10 minutes, and analyzed by Western blotting.

## Immunofluorescence assay

HEK293T cells were grown on confocal dishes for 24 hours, then transfected with different site mutant plasmids of N protein for 48 hours. Collected the above cells, removed the culture medium, and fixed the cells with 4% paraformaldehyde for 30 minutes, then permeabilized with PBS containing 0.1% TritonX-100 for 5 minutes. Finally, closed with PBS containing 5% BSA for 1 hour. Incubated the cells with the corresponding primary antibody overnight, and then staining with Cy3 and FITC coupled IgG secondary antibodies for 30 minutes. The nucleus were stained with DAPI for 5 minutes, washed with cold PBS 3 times, and observed under confocal fluorescence microscopy (Leica, TCS, SP8).

## Protein purification

To construct pET-28a-His-SARS-CoV-2-N$^{mut}$, mutant N gene was sub-cloned into pET-28a-His at BamHI and SacI sites. Plasmid pET-28a-His-N$^{mut}$ was transfected into Escherichia coli strain BL21. After growing in Kanamycin-resistant LB medium at 37°C until $OD_{600}$ reached 0.6–0.8, IPTG was added to a concentration of 0.2 mM, and medium was transferred to 16°C for 12–16 h. Cells were harvested and sonicated in lysis buffer (50 mM $NaH_2PO_4$, 100 mM PMSF, 300 mM NaCl, 10 mM imidazole, pH8.0). Lysates were centrifuged at 12000 rpm for 15 min to move debris. Supernatants were loaded into Ni-NTA Agarose columns (QIAGEN), shaken at 4°C for 2 h, and slowly flowed from columns and washed twice with wash buffer (50 mM $NaH_2PO_4$, 100 mM PMSF, 300 mM NaCl, 20 mM imidazole, pH8.0). Recombinant His-N$^{mut}$ protein was eluted using elution buffer (50 mM $NaH_2PO_4$, 100 mM PMSF, 300 mM NaCl, 250 mM imidazole, pH8.0). Recombinant His-N$^{mut}$ containing reduced imidazole was replaced into PBS using Millipore ultrafiltration tube. To construct pGEX-6p-1-GST-RIG-I, RIG-I gene was sub-cloned into pGEX-6p-1 at SmaI and XhoI sites. Plasmid pGEX-6p-1-GST-RIG-I was transfected into Escherichia coli strain BL21. After growing in Ampicillin-resistant LB medium at 37°C until $OD_{600}$ reached 0.6–0.8, IPTG was added to a concentration of 0.4 mM, and medium was transferred to 16°C for 16 h. Cells were harvested and sonicated in binding buffer (1 × PBS, pH7.4). Lysates were centrifuged at 12000 rpm for 15 min to move debris. Supernatants were loaded into GSH Magnetic Agarose Beads (Beyotime), shaken at 4°C for 2 h, and place on the magnetic frame and separate for 10s to remove the supernatant, then washed twice with Elution buffer (50 mM Tris-HCl, 10 mM GSH, pH8.0).

## RNA binding Protein Immunoprecipitation Assay (RIP)

HEK293T or Caco-2 cells were transfected with indicated plasmids for 24 hours. Then Caco-2 cells were infected with SARS-CoV-2 GFP/ΔN (MOI = 0.5) for another 48 hours. The above cells were collected and washed the cells twice with cold PBS. Resuspended the cells with RIP buffer plus RNasin. HEK293T cells were added the purified RNA of SARS-CoV-2 GFP/ΔN VLP (10 μg). After shaking incubation for 60 minutes, N proteins (mutant and WT) were immunoprecipitated with Flag-tag antibody and pulldown using Protein A/G Mag-Beads, washing four times by using the same RIP buffer and isolated RNAs were analyzed by RT-qPCR assay using specific viral N gene (ORF3a).

## Delta variant N protein mutated residue annotation

The structure of the SARS-CoV-2 Delta and Omicron variant N protein in this study were reconstructed using SWISS-MODEL Server based on original N protein model (PDB: 8fd5.1 in RCSB Protein Data Bank from earlier study. Mutated residues were analyzed and annotated using PyMOL 2.0.

## Statistical analyses

All experiments were repeated at least three times. The t-test was used to compare two groups and one-way ANOVA for multiple groups (GraphPad Prism7). The data were defined statistically significant when $P \leq 0.05$ (*), $P \leq 0.01$ (**) and $P \leq 0.001$ (***).

## Supporting information

**S1 Fig. A phylogenetic tree and site mutation construction of SARS-CoV-2 N protein. (A)** A phylogenetic tree constructed with N protein of different variants, with Bootstrap values greater than 50 is labeled on the branches. **(B–E)** HEK293T or Caco-2 cells were transfected with plasmids encoding Delta (B, D) or Omicron (C, E) mutated N protein for 24 h. The total RNA in the cells was extracted and the mRNA level of N protein was detected by RT-PCR. Cell lysates were analyzed by immunoblotting. Data are representative of three independent experiments and one representative is shown (B–E). Error bars indicate SD of technical triplicates. Values are mean ± SEM. *$P \leq 0.05$, **$P \leq 0.01$, ***$P \leq 0.001$, NS means No significant difference, two-tailed Student's t-test.
(TIF)

**S2 Fig. The R203M and D377Y mutation of N protein promote SARS-CoV-2-VLP production in HEK293T-ACE2 cells. (A)** HEK293T cells were stably infected with Lentivirus-CT or Lentivirus-ACE2, Cell lysates were analyzed by immunoblotting. **(B)** Caco-2 cells were stably infected with Lentivirus-CT or Lentivirus-N, Cell lysates were analyzed by immunoblotting. **(C)** Experimental scheme. HEK293T stable expression of ACE2 (HEK293T-ACE2) cells were transfected with plasmids encoding different mutated N protein for 24 h, and then infected with SARS-CoV-2 GFP/ΔN (MOI = 5) for 2 h, washed and incubated for an additional 48 h. The cell culture medium was collected to infect the Caco-2 stable expression of N (Caco-2-N) cells for 48 h. GFP fluorescence was observed by microscopy and viral RNA in supernatant was determined by RT-qPCR assay. **(D)** GFP expression was observed in HEK293T-ACE2 cells using microscopy at indicated time after inoculation. **(E)** The total RNA in supernatant of HEK293T-ACE2 cells was extracted and RT-qPCR assays were conducted to determine viral copies. Cell lysates were analyzed by immunoblotting. **(F)** GFP expression was observed in Caco-2-N cells using microscopy at indicated time after inoculation. **(G)** The total RNA in supernatant of Caco-2-N cells was extracted and RT-qPCR assays were conducted to determine viral copies. CT-Lev means CT- Lentivirus (A, B). ACE2-Lev means ACE2- Lentivirus (A). N-Lev means N- Lentivirus (B). Control means pcDNA3.1(+)-3 × flag empty plasmid (D–G). Data are representative of three independent experiments and one representative is shown. Error bars indicate SD of technical triplicates. Values are mean ± SEM. *$P \leq 0.05$, **$P \leq 0.01$, ***$P \leq 0.001$, two-tailed Student's t-test.
(TIF)

**S3 Fig. The R203M and D377Y mutation of N protein inhibited IFN-β production. (A)** HEK293T cells were transfected with IFN-β luciferase reporter pIFN-β-Luc, pPRL-TK and different mutated N protein for 24 h and then transfected with poly(I:C) (2 μg/mL) for 16 h. Cell lysates were harvested, IFN-β-Luc reporter activity was determined by dual luciferase

reporter assays. **(B)** HEK293T-ACE2 cells were transfected with plasmids encoding different mutated N protein for 24 h, and then infected with SARS-CoV-2 GFP/ΔN (MOI = 5) for 2 h, washed and incubated for an additional 48 h. The total RNA in cells was extracted and RT-qPCR assays were conducted to determine IFN-α, IFN-β and IFN-γ. **(C and D)** Caco-2 cells were transfected with plasmids encoding different mutated N protein for 24 h, and then infected with SARS-CoV-2 (MOI = 0.5) for 2 h, washed and incubated for an additional 48 h. The total RNA in cells was extracted and RT-qPCR assays were conducted to determine IFN-α (C) and IFN-γ (D). **(E)** HEK293T-ACE2 cells were transfected with plasmids encoding different mutated N protein for 24 h, and then infected with SARS-CoV-2 GFP/ΔN (MOI = 0.5) for 2 h, washed and incubated for an additional 48h. Supernatants were analyzed by ELISA for IFN-β. Control means pcDNA3.1(+)-3 × flag empty plasmid (A–E). Data are representative of three independent experiments and one representative is shown. Error bars indicate SD of technical triplicates. Values are mean ± SEM. *$P \leq 0.05$, **$P \leq 0.01$, ***$P \leq 0.001$, NS means No significant difference, two-tailed Student's t-test.
(TIF)

**S4 Fig. The R203M and D377Y mutation of N protein had no influence in interacting with PACT, G3BP1, MDA5 and MDA5-induced production of INF-β. (A–C)** HEK293T cells were co-transfected with plasmids encoding different mutated N protein and G3BP1 (A), PACT (B) or MDA5 (C) for 24 h. Cell lysates were immunoprecipitated using anti-Flag antibody, and analyzed using anti-Flag, anti-HA and anti-GAPDH antibody. Cell lysates (40 μg) was used as Input. **(D)** HEK293T cells were transfected with plasmids encoding different mutated N protein for 24 h and then infected with SeV (MOI = 0.1) for 16 h. Cell lysates were immunoprecipitated using anti-Flag antibody, and analyzed using anti-Flag, anti-RIG-I and anti-GAPDH antibody. Cell lysates (40 μg) was used as Input. **(E)** Coomassie blue staining analysis of the purified RIG-I-GST protein. **(F)** Coomassie blue staining analysis of the purified mutant N-His protein. **(G)** IFN-β luciferase reporter pIFN-β-Luc, pPRL-TK, different mutated N protein and RIG-I protein or pIFN-β-Luc, pPRL-TK, different mutated N protein and MDA5 protein for 24 h, and then infected with SeV (MOI = 0.1) for 16 h. Cell lysates were harvested, IFN-β-Luc reporter activity was determined by dual luciferase reporter assays. Control means pcDNA3.1(+)-3 × flag empty plasmid (A–D and G) or purified His-tagged empty protein (F). Data are representative of three independent experiments and one representative is shown.
(TIF)

**S5 Fig. The R203M and D377Y mutation of N protein inhibited the colocalization with RIG-I.** Caco-2 cells were co-transfected with plasmids encoding different mutated N protein and HA-tagged RIG-I protein for 24 h and then infected with SARS-CoV-2 GFP/ΔN (MOI = 5) for 48 h. Nucleus marker DAPI (blue), HA-tagged RIG-I protein (yellow), GFP- SARS-CoV-2 GFP/ΔN (green) and Flag- mutated N protein (red) were then visualized with confocal microscopy. Data are representative of three independent experiments and one representative is shown.
(TIF)

**S6 Fig. The R203M and D377Y mutation of N protein had no influence in the interaction between TRIM25 and RIG-I and TRIM25-mediated ubiquitination of RIG-I. (A)** HEK293T cells were co-transfected with plasmids encoding different mutated N protein with RIG-I and TRIM25 for 24 h. Cell lysates were immunoprecipitated using anti-HA antibody, and analyzed using anti-Flag, anti-HA, anti-Myc and anti-GAPDH antibody. Cell lysates (40 μg) was used as Input. **(B)** HEK293T cells were co-transfected with HA-RIG-I,

Myc-TRIM25, different mutated N protein and Myc-Ubiquitin for 24 h. Cell lysates were immunoprecipitated using anti-HA antibody, and analyzed using anti-Ub, anti-HA, anti-GAPDH and anti-TRIM25 antibody. Cell lysates (40 μg) was used as Input. Control means pcDNA3.1(+)-3 × flag empty plasmid. Data are representative of three independent experiments and one representative is shown.
(TIF)

**S7 Fig. The experimental scheme of RNA Binding Protein Immunoprecipitation Assay (RIP). (A)** HEK293T cells were transfected with plasmids encoding different mutated N protein for 24 h, Cell lysates were collected and incubated with purified RNA of SARS-CoV-2 GFP/ΔN VLP (10 μg) for 60 min, then the mixture was pulldown using Protein A/G Mag-Beads, washing and isolated RNAs were analyzed by RT-qPCR assay using specific viral N gene. **(B)** Caco-2 cells were transfected with plasmids encoding different mutated N protein for 24 h and then infected with SARS-CoV-2 GFP/ΔN (MOI = 0.5) for 2 h, washed and incubated for an additional 48 h. Cell lysates were pulldown using Protein A/G MagBeads, washing and isolated RNAs were analyzed by RT-qPCR assay using specific viral N gene. **(C)** Caco-2 cells were transfected with plasmids encoding different mutated N protein plus RIG-I protein for 24 h, Cell lysates were collected and incubated with purified RNA of SARS-CoV-2 GFP/ΔN VLP (10 μg) for 60 min, then the mixture was pulldown using Protein A/G MagBeads, washing and isolated RNAs were analyzed by RT-qPCR assay using specific viral N gene. **(D)** Caco-2 cells were transfected with plasmids encoding different mutated N protein plus RIG-I protein for 24 h and then infected with SARS-CoV-2 GFP/ΔN (MOI = 0.5) for 2 h, washed and incubated for an additional 48 h. Cell lysates were pulldown using Protein A/G MagBeads, washing and isolated RNAs were analyzed by RT-qPCR assay using specific viral N gene.
(TIF)

**S1 Table. qRT-PCR primers used in this study.**
(DOCX)

**S2 Table. PCR primers used in this study.**
(DOCX)

**S1 File. Unprocessed images and all raw data.**
(DOCX)

## Acknowledgments

We are grateful for the support and assistance provided by Professor Jianguo Wu, who has passed away, for this research.

## Author contributions

**Conceptualization:** Yongkui Li, Pan Pan.

**Data curation:** Feng Liao, Junxian Ou.

**Formal analysis:** Yongkui Li, Moran Li, Heng Xiao, Feng Liao, Miaomiao Shen.

**Funding acquisition:** Yongkui Li, Qiwei Zhang, Pan Pan.

**Investigation:** Yongkui Li, Moran Li, Heng Xiao, Feng Liao, Miaomiao Shen, Weiwei Ge, Junxian Ou, Yuqing Liu, Lumiao Chen, Yue Zhao, Pin Wan, Jinbiao Liu, Jun Chen, Xianwu Lan, Shaorong Wu, Qiang Ding, Geng Li, Qiwei Zhang.

**Methodology:** Yongkui Li, Moran Li, Heng Xiao, Feng Liao, Miaomiao Shen, Geng Li, Qiwei Zhang, Pan Pan.

**Project administration:** Pan Pan.

**Resources:** Yongkui Li, Moran Li, Heng Xiao, Feng Liao, Miaomiao Shen, Geng Li, Qiwei Zhang, Pan Pan.

**Software:** Feng Liao, Junxian Ou.

**Supervision:** Yongkui Li, Pan Pan.

**Validation:** Yongkui Li, Moran Li, Heng Xiao, Feng Liao, Miaomiao Shen, Geng Li, Qiwei Zhang, Pan Pan.

**Visualization:** Yongkui Li, Moran Li, Heng Xiao, Feng Liao, Miaomiao Shen, Weiwei Ge, Junxian Ou, Yuqing Liu, Lumiao Chen, Yue Zhao, Pin Wan, Jinbiao Liu, Jun Chen, Xianwu Lan, Shaorong Wu, Qiang Ding, Geng Li, Qiwei Zhang, Pan Pan.

**Writing – original draft:** Yongkui Li, Moran Li, Qiwei Zhang, Pan Pan.

**Writing – review & editing:** Yongkui Li, Qiwei Zhang, Pan Pan.

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
