## [Decision Letter · Decision Letter 0]

18 Sep 2024

Dear Dr Pan,

Thank you very much for submitting your manuscript "The R203M and D377Y Mutations of N Protein promote SARS-CoV-2 Infectivity through impairing RIG-I-mediated antiviral signaling" for consideration at PLOS Pathogens. As with all papers reviewed by the journal, your manuscript was reviewed by members of the editorial board and by several independent reviewers. In light of the reviews (below this email), we would like to invite the resubmission of a significantly-revised version that takes into account the reviewers' comments.

We cannot make any decision about publication until we have seen the revised manuscript and your response to the reviewers' comments. Your revised manuscript is also likely to be sent to reviewers for further evaluation.

Sincerely,

Victor Robert DeFilippis

Guest Editor

PLOS Pathogens

Sonja Best

Section Editor

PLOS Pathogens

Michael Malim

Editor-in-Chief

PLOS Pathogens

orcid.org/0000-0002-7699-2064

Reviewer's Responses to Questions

**Part I - Summary**

Reviewer #1: In this manuscript the authors assessed the impact of two unique mutations that were identified in the N protein of the Delta variant of SARS-CoV-2. During the pandemic, several mutations in N protein have been observed in the different variants. This group used sequence analysis to identify those mutations and then used several genetic techniques to assess the impact of these mutations on both virus replication and interferon production. They compared the impact of these mutations, both singly and in combination, with WT virus as well as with mutations that were identified in the Omicron variant. They concluded that the Delta variant specific mutations promoted SARS-CoV-2 infection and had increased inhibitory activity against RIG-I-dependent IFN production, and thus resulted in reduced levels of IFN following infection with SeV, SARS-CoV-2, or treatment with polyI:C. They then claim that these mutations enhance the affinity of N protein for viral RNA which reduces the ability of RIG-I to interact with viral RNA. Finally, they used AAV to express different N protein mutants in mice and then infect them with VSV, a virus highly sensitive to IFN. They found that the Delta variant N protein and specifically the R203M/D377Y mutant N protein inhibited the production of IFN in mice which led to increased VSV replication and enhanced disease.

Defining the relative importance of variant mutations of SARS-CoV-2 in its ability to replicate and cause disease are critical to understanding how best to target specific variants for treatment and to help identify new variants that may cause increased disease. The authors here have convincingly showed that these two primary N protein mutations in the Delta variant enhance the ability of N protein to promote virus replication, bind RNA, and further enhance N protein’s ability to repress IFN responses. The demonstration that these mutations enhance VSV replication in vivo (for which N protein has no role in replication) strongly demonstrates its IFN inhibitory activity. However, there are several issues with the interpretation that the mechanism of IFN repression is due to reduced RIG-I-RNA interaction. Furthermore, no data was provided that demonstrated that the enhanced SARS-CoV-2 replication is due to reduced IFN production. Thus, the results do not fully support the conclusion stated in the title. Finally, the writing in the paper is not up to the standards of the journal, and thus the paper needs to be extensively edited before it is suitable for publication.

Reviewer #2: Overall, the paper confirms two critical amino acids in the N protein of SARS-CoV-2, where their combination mutant exhibits a dual function: suppressing the RIG-I-mediated interferon production, and enhancing viral replication through increasing RNA binding to the N protein. This discovery sheds light on the mechanisms behind SARS-CoV-2’s variability in viral RNA replication and its capacity to modulate host immune responses.

Reviewer #3: The R203M and D377Y Mutations of N Protein promote SARS-CoV-2 Infectivity through impairing RIG-I-mediated antiviral signaling

PLOS Pathogens

Summary:

The manuscript is discussing the SARS-CoV-2 nucleocapsid mutations that are found in certain variants. These mutations found in the Delta variants are compared to those found in Omicron variants. The authors claim that the N protein mutations found in the Delta variants promoted viral replication. Increase in viral replication caused by N protein mutations in Delta caused inhibition of interaction between RIG-I and MAVS that lead to increased viral replication. Induction of interferons was measured after infection with different interferon activators and to original SARS-CoV-2 virus indicting that the N protein mutant inhibit IFN-Beta but no inhibition of other IFN-types of . Authors investigated the role of the N protein mutations on interacting with other cellular proteins that are involved in activating interferons. N protein with specific mutations inhibited interaction with intracellular RIG-I protein that is activated by binding to viral RNA. Furthermore, the RIG-I and MAVS interaction was inhibited by these N mutations leading to inhibition of phosphorylation of TBK1 and IRF3 proteins. Nucleocapsid mutations phenotypes were also investigated in C57BL/6 mice using AAV system and activation of RIG-I pathway using VSV. The manuscript is well written and investigated SARS-CoV-2 some nucleocapsid mutations with intensive experiments.

**Part II – Major Issues: Key Experiments Required for Acceptance**

Reviewer #1: 1. The paper has many grammatical errors, incorrect word usage, incorrect statements etc. For example, line 77 indicates that N protein helps form cellular stress granules, when it actually prevents their formation. At multiple places the authors stated that the omicron N protein promoted IFN production compared to the delta strain N protein (i.e. line 190), but in reality, the N protein from omicron represses IFN production, just not to the same extent. The correct terminology is that the omicron N protein had reduced inhibition of IFN production as compared to the delta N protein. Another example is line 163, where it states that the omicron N protein inhibited trVLP replication compared to delta mutant N protein, but again it just promotes virus replication to a lesser degree. The manuscript also lacked references in many places (i.e. line 214) and the title is not well written and has some words have capital letters where there shouldn’t be. The bottom line is that the manuscript requires extensive editing.

2. The major issue in this manuscript is that the data supporting their final model, where the delta N protein has enhanced RNA binding, which results in reduced RIG-I-RNA interactions and reduced IFN production does not hold up when comparing all of the mutants. Most notably, the omicron mutant protein RG203/204KR reduced RIG-I-RNA binding to the same degree as the Delta mutant protein (Fig. 6) but did not have the same reduction in IFN production following SeV, SARS-CoV-2, or polyI:C treatment (Figs. 4, 5). It also had roughly similar levels of RIG-I binding (Fig. 5B) and promoted virus replication to the same degree as the WT protein (Figs 2-3), further indicating that reduced RIG-I-RNA binding does not impact IFN or virus replication. Furthermore, the authors never analyzed the impact of N protein on MDA5-mediated induction of IFN, which is important as MDA5 is also a major RNA sensor during CoV infection and poly(I:C) treatment. Though it appears that since the N protein reduced IFN production from SeV and CoV infection, that it probably inhibits IFN production induced from either sensor.

3. In line with the previous comment, it was odd that the Delta N protein had reduced RIG-I binding. Has it not been shown that the interaction of N protein with RIG-I is critical for its ability to repress IFN production? Or is this unknown? Perhaps they only interact through their ability to bind RNA, which might help explain the data. The authors should perform Co-IPs with N protein and RIG-I in the presence of RNase (or at least in the absence of RNA) to determine if the interaction between the two proteins is dependent on RNA binding. They have recombinant proteins and could easily do this in vitro. Others have shown that N protein interacts with TRIM25 which reduces the ubiquitination of RIG-I. It would be prudent to also determine if the Delta N protein impacts RIG-I-TRIM25 interactions and its ubquitination. (PMIDs:33801464; 32295922; 28148787)

4. There was no data to support the conclusion that “R203M and D377Y Mutations of N Protein promote SARS-CoV-2 Infectivity through impairing RIG-I-mediated antiviral signaling”. The authors did not perform any experiment that specifically demonstrated that IFN signaling impacted virus replication in these cells. This would be required to make this conclusion.

Reviewer #2: Generally, there should be no more than 3 such required experiments or major modifications for a "Major Revision" recommendation. If more than 3 experiments are necessary to validate the study conclusions, then you are encouraged to recommend "Reject".

Three new experiments are needed: 1) compare the viral growth kinetic of the original strain, the Delta and the Omicron variants in Calu-3 cells or Caco-2 cells. After infection, at various time points post-infection (eg. 6, 12, 24, 48 and 72 hours) collect supernatants and measure viral titers to quantify infectious particles. At these time points, the authors should detect the type I interferon production by qPCR or ELISA to assess the differences in IFN production induced by these three strains; 2) In figure 5 E, a notable observation is that IRF3 was not detected in all cells especially in control group. The experiment needs further investigation.

Reviewer #3: (No Response)

**Part III – Minor Issues: Editorial and Data Presentation Modifications**

Reviewer #1: (No Response)

Reviewer #2: Minor modifications: 1) SARS-CoV-2 has a complex genome that includes several open reading frames (ORFs) with key roles in viral pathogenesis. Notably, ORF3, ORF6, and ORF9b are critical for antagonizing the host's innate immune response. These ORFs interfere with the host’s antiviral defenses by disrupting interferon signaling and other immune mechanisms, thereby facilitating viral persistence and replication. Among these, ORF6 has been identified as crucial for viral replication. Particularly noteworthy is the D61L mutation in ORF6, which emerged in the Omicron strains. This mutation impairs immune evasion, highlighting its essential role in the virus’s ability to propagate. Please update introduction section with these specific details; 2) The structure of various N protein variants is illustrated in figure 1B. This figure only presents the mutant sites of N protein but does not provide a detailed description of the differences between these variant structures; 3) The legends for figure 2B and 2D are currently unclear and do not provide a detailed description. The figure legends for figure 2B and 2D, as well as for Figure 2C and 2E, are identical and do not clearly describe the differences between the two experiments; 4) In figure 4, it was observed that mutant of N protein inhibits the production of interferon-beta but does not affect interferon-alpha production. Please discuss the reasons for this selective inhibition in Discussion section; 5) The manuscript contains many awkward/incorrect uses of words and phrases. The authors need to use a science editing service that employs native English speakers, as these problems make the manuscript difficult to read.

Reviewer #3: Here are some notes to the authors to address before resubmitting the manuscript.

Comments:

1. There are a couple of studies about SARS-CoV-2 detect a role of IFN type III. Could you elaborate of not measuring type III IFN beside the other interferon in your study?

2. I would like to know what strain you refer to by original SARS-CoV-2 strain, and its accession number. (e.g. Wuhan strain)

3. Figure 2A, the first part of “de novo infection” of VLP is not accurate as stated in the protocol and in the text. Needs more clarifications or modifications to fit with the procedure in the protocol.

4. Fig 2D, it would be better to write the time of infection in the figure.

5. Figure 3A, the first part of “de novo infection” of SARS-CoV-2 is not accurate as stated in the protocol and in the text. Infection of original SARS-CoV-2 was after transfecting the cells with N and N-mutant. This needs more clarifications or modifications to fit with the procedure in the protocol.

6. I would suggest that the viral copies per mL to be graphed as Log scale to visual the difference between mutants easily.

7. Line 171, “The SARS-CoV-2 in supernatant continued to infect Caco-2 cells…” that is a second infection?!

8. Line 186, “Sev” I assume you refer to Sendai virus, you should state that before the acronym.

9. I would suggest that the SARS-CoV-2-trVLP work should be sectioned in 2 parts in the text. The SARS-CoV-2-trVLP work on Caco-2 cells should be separated from the HEK293T-ACE2 cells as these cells have been used as a confirmatory data and placed in the supplementary data section. (Section start from line 137 to 165)

10. Line 224, “co-location” should be “co-localization”

11. Fig 6E right, it would be better to label the proteins as in the other figures. I would probably move this figure to the supplementary file.

12. Line 287, “(incubation with the same quality of RIG-I protein and viral RNA …), you mean quantity (ug) not quality.!?

13. Since the phenotype of Omicron variant N mutation at P13L behave different when combined with R203G/K204R, can you elaborate on the function of the P13L mutation in Omicron variant.

14. Figure 8 that summarizes the phenotype of Delta and Omicron nucleocapsid variants is not clear enough, could you please redraw that figure with more clarification indicating what is inhibited and what is increased to show differences between the 2 variants.

15. I know how difficult it is to make infectious clones with specific mutations in the nucleocapsid, but the conclusion of the results should exclude that the phenotype observed in this manuscript using trVLP system would be similar to the phenotype of mutations using actual infectious SARS-CoV-2 clones harbouring the same mutations. These comments should be stated very clearly in the discussion, especially for interpreting mice data.

PLOS authors have the option to publish the peer review history of their article (what does this mean? ). If published, this will include your full peer review and any attached files.

**Do you want your identity to be public for this peer review?** For information about this choice, including consent withdrawal, please see our Privacy Policy .

Reviewer #1: No

Reviewer #2: No

Reviewer #3: No

Figure Files:

Data Requirements:

Please note that, as a condition of publication, PLOS' data policy requires that you make available all data used to draw the conclusions outlined in your manuscript. Data must be deposited in an appropriate repository, included within the body of the manuscript, or uploaded as supporting information. This includes all numerical values that were used to generate graphs, histograms etc.. For an example see here on PLOS Biology: http://www.plosbiology.org/article/info%3Adoi%2F10.1371%2Fjournal.pbio.1001908#s5 .
---

## [Decision Letter · Decision Letter 1]

1 Jan 2025

PPATHOGENS-D-24-01512R1

The R203M and D377Y mutations of the nucleocapsid protein promote SARS-CoV-2 infectivity by impairing RIG-I-mediated antiviral signaling

PLOS Pathogens

Dear Dr. Pan,

Thank you for submitting your manuscript to PLOS Pathogens. After careful consideration, the reviewers feel that the critiques were appropriately addressed and the revised manuscript is greatly improved. However, some minor changes are necessary before full acceptance. Therefore, we invite you to submit a revised version of the manuscript that addresses the points raised during the review process.

Please submit your revised manuscript within 30 days Mar 02 2025 11:59PM. If you will need more time than this to complete your revisions, please reply to this message or contact the journal office at plospathogens@plos.org. Please include the following items when submitting your revised manuscript:

We look forward to receiving your revised manuscript.

Kind regards,

Victor Robert DeFilippis

Guest Editor

PLOS Pathogens

Sonja Best

Section Editor

PLOS Pathogens

Sumita Bhaduri-McIntosh

Editor-in-Chief

PLOS Pathogens

orcid.org/0000-0003-2946-9497

Michael Malim

Editor-in-Chief

PLOS Pathogens

orcid.org/0000-0002-7699-2064

**Journal Requirements:**

1) Please note that the Funding information and the Authors' Contributions should not be uploaded as a separate file in the online submission form. Please remove it and make sure that only those relevant to the current version of the manuscript are included.

2) We note that your Data Availability Statement is currently as follows: "All relevant data are within the manuscript and its Supporting Information files." Please confirm at this time whether or not your submission contains all raw data required to replicate the results of your study. Authors must share the “minimal data set” for their submission. PLOS defines the minimal data set to consist of the data required to replicate all study findings reported in the article, as well as related metadata and methods (https://journals.plos.org/plosone/s/data-availability#loc-minimal-data-set-definition ).

3) Please amend your detailed Financial Disclosure statement. This is published with the article. It must therefore be completed in full sentences and contain the exact wording you wish to be published.

4) Your current Financial Disclosure states, "This work was supported by the National Natural Science Foundation of China (32200117 to P.P. and 92269103 to Q.Z.). Open Research Fund Program of the State Key Laboratory of Virology of China (2021KF003 to P.P.). Open Research Fund Program of Guangdong Provincial Key Laboratory of Virology (2022KF003 to P.P.). R&D Program of Guangzhou Laboratory (SRPG22-006 to Q.Z.). Guangdong Basic and Applied Basic Research Foundation (2024A1515011433 to Y.L. and 2024A1515013063 to P.P), and Fundamental Research Funds for the Central Universities (21623404 to Y.L. and 21623222 to P.P), Guangzhou Science and Technology Plan Project (Youth Doctor "Setting Sail", 2024A04J4102 to P.P)."

However, your funding information on the submission form indicates receiving only two grants. Please ensure that the funders and grant numbers match between the Financial Disclosure field and the Funding Information tab in your submission form. Note that the funders must be provided in the same order in both places as well.

Please indicate by return email the full and correct funding information for your study and confirm the order in which funding contributions should appear. Please be sure to indicate whether the funders played any role in the study design, data collection and analysis, decision to publish, or preparation of the manuscript.

**Reviewers' Comments:**

Reviewer's Responses to Questions

**Part I - Summary**

Reviewer #1: The authors have addressed all of my concerns, having done some really nice new experiments to better prove their overall conclusions.

There are just a couple of minor grammatical changes yet to be made.

Line 31. This should be "Viral protein mutations......"

Line 294. "inhibited the direct interaction"

Line 296. Should be Fig. 5B

Line 307. "both N protein mutations R203M and D377Y"

Line 380. The first sentence of this paragraph is incomplete.

Reviewer #2: In this manuscript, the authors investigated the functional impact of two key amino acids mutations in the N protein of the SARS-CoV-2 Delta variant. Compare to omicron variant, the N protein mutations in the Delta variant enhanced viral replication. Mechanistic studies revealed that the N mutants bound to RIG-I, suppressing interferon production and evading the host immune response. Furthermore, the N protein mutants exhibited enhanced RNA-binding capability, directly promoting viral replication. These findings illuminate the mechanism underlying SARS-CoV-2’s variability in viral RNA replication and its ability to modulate host immune responses.

Reviewer #3: The modification made to this manuscript is sufficient for publication

**Part II – Major Issues: Key Experiments Required for Acceptance**

Reviewer #1: (No Response)

Reviewer #2: I have read the response to the reviewer's comments and I am satisfied that the authors have made reasonable efforts to address these comments. A far as I am concerned they have answered all my queries and I think the manuscript is much improved.

Reviewer #3: (No Response)

**Part III – Minor Issues: Editorial and Data Presentation Modifications**

Reviewer #1: (No Response)

Reviewer #2: All of my original comments have been addressed.

Reviewer #3: (No Response)

PLOS authors have the option to publish the peer review history of their article (what does this mean? ). If published, this will include your full peer review and any attached files.

**Do you want your identity to be public for this peer review?** For information about this choice, including consent withdrawal, please see our Privacy Policy .

Reviewer #1: No

Reviewer #2: No

Reviewer #3: No

**Figure resubmission:**

While revising your submission, please upload your figure files to the Preflight Analysis and Conversion Engine (PACE) digital diagnostic tool, https://pacev2.apexcovantage.com/ . PACE helps ensure that figures meet PLOS requirements. To use PACE, you must first register as a user. Registration is free. Then, login and navigate to the UPLOAD tab, where you will find detailed instructions on how to use the tool. If you encounter any issues or have any questions when using PACE, please email PLOS at figures@plos.org. Please note that Supporting Information files do not need this step. If there are other versions of figure files still present in your submission file inventory at resubmission, please replace them with the PACE-processed versions.
---

## [Editor Report · Decision Letter 2]

3 Jan 2025

Dear Dr Pan,

We are pleased to inform you that your manuscript 'The R203M and D377Y mutations of the nucleocapsid protein promote SARS-CoV-2 infectivity by impairing RIG-I-mediated antiviral signaling' has been provisionally accepted for publication in PLOS Pathogens.

Best regards,

Victor Robert DeFilippis

Guest Editor

PLOS Pathogens

Sonja Best

Section Editor

PLOS Pathogens

Sumita Bhaduri-McIntosh

Editor-in-Chief

PLOS Pathogens

orcid.org/0000-0003-2946-9497

Michael Malim

Editor-in-Chief

PLOS Pathogens

orcid.org/0000-0002-7699-2064
---

## [Editor Report · Acceptance letter]

Dear Dr Pan,

We are delighted to inform you that your manuscript, "The R203M and D377Y mutations of the nucleocapsid protein promote SARS-CoV-2 infectivity by impairing RIG-I-mediated antiviral signaling," has been formally accepted for publication in PLOS Pathogens.

Best regards,

Sumita Bhaduri-McIntosh

Editor-in-Chief

PLOS Pathogens

orcid.org/0000-0003-2946-9497

Michael Malim

Editor-in-Chief

PLOS Pathogens

orcid.org/0000-0002-7699-2064